



# On the Conceptual Complexity of Non-Point Source Management: Impact of Spatial Variability

Christopher Vincent Henri[1], Thomas Harter[1], and Efstathios Diamantopoulos[2]

[1]University of California, Davis, Center for Watershed Sciences, Veihmeyer Hall, Davis, CA 95616, USA.
[2]University of Copenhagen, Department of Plant and Environmental Sciences, Thorvaldsensvej 40, Copenhagen, DK-1871, Denmark.

**Correspondence:** Christopher Vincent Henri (chenri@ucdavis.edu)

**Abstract.** Non-point source (NPS) pollution has degraded groundwater quality of unconsolidated sedimentary basins over many decades. Properly conceptualizing NPS pollution from the well scale to the regional scale leads to complex and expensive numerical models: Key controlling factors of NPS pollution - recharge rate, leakage of pollutants, and soil and aquifer hydraulic properties - are spatially and, for recharge and pollutant leakage, temporally variable. This leads to high uncertainty
in predicting well pollution. On the other hand, concentration levels of some key NPS contaminants (salinity, nitrate) vary within a limited range (<2 orders of magnitude); and significant mixing occurs across the aquifer profile along the most critical compliance surface: drinking water wells with their extended vertical screen length. Here, we investigate, whether these two unique NPS contamination conditions reduce uncertainty such that simplified spatiotemporal representation of recharge and contaminant leakage rates and of hydraulic conductivity are justified when modeling NPS pollution. We employ a Monte
Carlo-based stochastic framework to assess the impact of model homogenization on key management metrics for NPS contamination. Results indicate that travel time distributions are relatively insensitive to the spatial variability of recharge and contaminant loading, while capture zone and contaminant time series exhibit some sensitivity to source variability. In contrast, homogenization of aquifer heterogeneity significantly affects the uncertainty assessment of travel times and capture zone delineation. Surprisingly, regional statistics of well concentration time series are fairly well reproduced by a series of equivalent
homogeneous aquifers, highlighting the role of NPS solute mixing along well screens.

## 1 Introduction

The use of agrochemicals to address an ever growing food demand has led to the contamination of many sedimentary groundwater basins underlying intensively farmed regions (Nolan et al., 2002; Zektser and Everett, 2004; Rockstrom et al., 2009).
Given the broad, continuous expanse of agricultural pollution sources across affected groundwater basins, this type of large
scale pollution is often referred to as non-point source (NPS) pollution (Ritter and Shirmohammadi, 2000). The development of effective protection or remediation strategies in groundwater bodies affected by NPS pollution will require understanding of the dynamics of NPS pollution in groundwater systems. Important aspects of NPS pollution are pollutant travel times, the location of well source areas (capture zones) to identify specific pollution sources, and the long-term evolution of contaminant




levels in and across affected wells and streams. The predictive modeling of these processes and associated management metrics

is challenged by the inherent complexity of NPS pollution in groundwater systems.

Spatial variability represents a key source of complexity to be considered in understanding pollutant transport in the subsurface. Decades of investigation at contaminated industrial sites have highlighted the critical role that aquifer heterogeneity (e.g., the hydraulic conductivity) has in accurately understanding the solute transport behavior, to identify polluters, and to design effective remediation schemes and assess associated risk (e.g., Dagan and Nguyen, 1989; Cvetkovic et al., 1992; de Barros and

Nowak, 2010; Henri et al., 2016). Large aquifer heterogeneity significantly affects the macro-dispersive behavior of contaminant plumes emanating from point sources. Lacking data to characterize subsurface properties in sufficient detail introduces significant uncertainty in the prediction of solute transport, the design of remediation measures, and the prediction of future concentrations at wells of interest (e.g., Dagan, 1984; Rubin, 2003). The prediction of solute transport from the NPS to a compliance area of interest (e.g., extraction or observation wells) has been shown to be critically impacted by aquifer heterogeneity,

but also by mixing along the screen of production wells: contaminant mass arrivals in extraction wells can occur over decades to centuries and are characterized by significant uncertainty (Hua and Harter, 2006; Henri and Harter, 2019).

Unique to nonpoint sources, the spatial (and temporal) variability of the source itself across a groundwater basin introduces an additional level of system complexity. NPS pollution of groundwater is typically associated with dissolved solutes associated with groundwater recharge across the landscape. Both, recharge rates and contaminant concentrations in nonpoint sources are

subject to large spatial and temporal variability. The variability is partly due to spatially variable soil properties (e.g., Nielsen et al., 1973). These properties control infiltration, recharge to groundwater, and the fate and transport of contaminants in the unsaturated zone (Hillel, 1980). Landscape management that leads to NPS pollution releases, e.g., irrigation, fertilization, construction and maintenance of urban, domestic, and other distributed waste systems leaking incidentally or intentionally into groundwater, is also subject to large spatial and temporal variability (Jordan et al., 1997). As with aquifer properties, the

minutia of such spatial and temporal variability cannot be measured (or estimated) except at larger scales. For example, to the degree that differences exist in average recharge and pollutant loading between mappable landscape management systems, these may be explicitly represented in space and time (e.g., Loague and Corwin, 1998; Nolan et al., 2018). This includes NPS differences between different farming systems (Kladivko et al., 2004) or between crops (Logsdon et al., 2002). Similarly, the degree to which mappable soil units affect recharge and pollutant fate and transport to the water table can also be explicitly

represented (Biggar and Nielsen, 1976). However, spatial variability at smaller spatial scales or between individual units of the same mappable class are subject to stochastic variability (Sisson and Wierenga, 1981; Vereecken et al., 2007). Furthermore, both, the timing and the spatial distribution of mappable and smaller scale unknown landscape processes is a stochastic process from a regional management perspective, which is concerned with pollution dynamics across an ensemble of wells.

The dual complexity of aquifer heterogeneity and spatio-temporal source variability represent a largely unexplored chal-

lenge in the assessment and management of NPS pollution in aquifers. Yet, conceptually simplified approaches have been successfully employed to predict general trends and expected (average) contaminant behavior across ensembles of pollutant receptors of interest (wells, stream reaches) (e.g., Conan et al., 2003). Typically, these assessments lack any measures to also assess predictive uncertainties.





Some key characteristics of NPS contamination on the other hand make the NPS pollution system in groundwater well suited
for upscaling without loss of information relevant to understanding the range of impacts on receptors: First, the individual
compliance surface of interest (the groundwater-well interface, the groundwater-stream reach interface) is subject to complete
mixing prior to exposure (extracted well water, stream reach baseflow contribution). For example, production wells for urban
water supplies are typically screened over dozens of meters (Henri and Harter, 2019). Even domestic wells are typically
screened vertically over several meters of an aquifer system (Horn and Harter, 2009; Perrone and Jasechko, 2019). Similarly,
stream reaches mix across an aquifer area of several tens to tens of thousands of square meters. The source area associated with
such significantly sized compliance surfaces typically has length scales exceeding 100 m and frequently exceeding 1km (Horn
and Harter, 2009; Henri and Harter, 2019). As a result, extracted water will be a mixture of groundwater age and source water
quality (Weissmann et al., 2002; Koh et al., 2018).

Secondly, while the source is wide-spread, compliance levels of key NPS contaminants (e.g., salt, nitrate) are commonly
much less than one order of magnitude lower than the concentration in NPS recharge. This is characteristically different
from most point-source contamination, where concentrations at the source may exceed compliance levels by many orders of
magnitude (e.g., Frind et al., 1999). With the smaller difference between compliance and NPS recharge concentration, mixing
at the compliance surface (i.e., in the well screen, at the stream reach scale) acts to homogenize the NPS recharge signature
both, in space and time, thus reducing the need to accurately characterize the variability in space or in time to determine the
mixed concentration at an individual compliance surface (Kourakos et al., 2012). In contrast to assessing point sources of
industrial pollution, significant simplification in the spatiotemporal representation of, both or either, water and contaminant
leakage rates and hydraulic conductivity may be possible without loss of accuracy. Bastani and Harter (submitted) explored
the homogenization of temporal variability in NPS behavior. Yet, little work has been done to better understand and quantify
the degree to which spatio-temporal variability in NPS representation or spatial variability in aquifer representation can be
homogenized in NPS simulation tools while still accurately predicting NPS management metrics, including concentrations at
compliance surfaces.

In this paper, we assess the degree to which detailed spatial representation of both, the aquifer hydraulic conductivity and
of contaminant source parameters - recharge rate of water and contaminant loading from the NPS to the groundwater table -
can be homogenized in NPS models without reducing model accuracy. We consider three NPS management metrics and use a
comparative simulation approach for our assessment.

Our starting point is a set of simulations that predict the long term contamination of an aquifer from NPS pollution under
highly resolved heterogeneous aquifer and NPS source conditions. Results are compared against various simulation scenarios
with homogenized representations of the aquifer and source heterogeneity. We compare results from various homogenization
scenarios by focusing on three stochastic management metrics: the travel time distribution to production wells, the stochastic
capture zone, and the stochastic contaminant time series in well water. Assuming ergodicity (Dagan, 1990), stochastic man-
agement metrics are quantified both, for the pollution variability across an ensemble of production wells encountered over a
basin, and for the uncertainty about pollution levels at an individual well.





**Table 1.** Proportion and hydraulic conductivity of the four categories (g: gravel; s: sand; ms: muddy sand; m: mud)

|                             | g     | s    | ms   | m    |
| --------------------------- | ----- | ---- | ---- | ---- |
| proportion [%]              | 0.10  | 0.35 | 0.25 | 0.3  |
| hydraulic conductivity [m/d]| 200.0 | 50.0 | 0.5  | 0.01 |

## 2  Methodology

### 2.1  Reference case

We consider an unconsolidated sedimentary aquifer sytem typical of the Central Valley (California, USA), initially uncontaminated (e.g., pre-development state) and subject to nitrate pollution from agricultural NPS sources. The sub-region is characterized by a semi-arid Mediterranean climate, with dry summers and significant winter precipitation occurring mostly via the surrounding mountain ranges. The Central Valley groundwater basin is subject to intensive irrigated agricultural activities supported by reservoirs managing surface water inflows from surrounding mountain ranges and by groundwater. Over the past

eight decades, irrigation and groundwater pumping added a significant vertical flow component: Lateral groundwater flow, fed by mountain front recharge and discharged along the thalweg used to dominate the groundwater system dynamic. Modern groundwater discharge is mostly due to groundwater extraction. Recharge from intensive irrigation is superimposed on a weak lateral gradient, significantly increasing the importance of downward flows (Faunt, 2009). Water recharged from the irrigated landscape to groundwater bodies carries significant loading of agricultural NPS pollutants, such as salt or nitrate (e.g., Baram

et al., 2016).

The simulated soil and aquifer contamination setting represents conditions typically encountered in Central Valley's agricultural basins, but are not specific to a particular location. We represent heterogeneity in the hydraulic conductivity as well as the spatial variability in soil types and land use. The latter two are key characteristics that control spatial variability in recharge and contaminant leakage rates. The transfer of water and nitrate from the soil surface to the aquifer is estimated through the

modeling of flow and transport in the unsaturated zone for a series of typical soil types and crops found in the Central Valley.

#### 2.1.1  Representation of aquifer and soil heterogeneity

#### 2.1.2  Stochastic analysis

Uncertainty in the representation of the spatial variability of the aquifer and soil hydraulic conductivity is systematically accounted for through the use of a geostatistical model in a Monte Carlo framework (Rubin, 2003). The propagation of variability

and uncertainty into management metrics is assessed across an ensemble of production wells. Assuming ergodicity (Dagan, 1990), stochastic analysis is applied to first quantify uncertainty about pollution outcomes at individual wells and to secondly quantify regional spatial variability in pollution outcomes across an ensemble of wells: To characterize the uncertainty at an individual well, a large number of realizations of individual wells with equiprobable aquifer and soil realizations is generated.





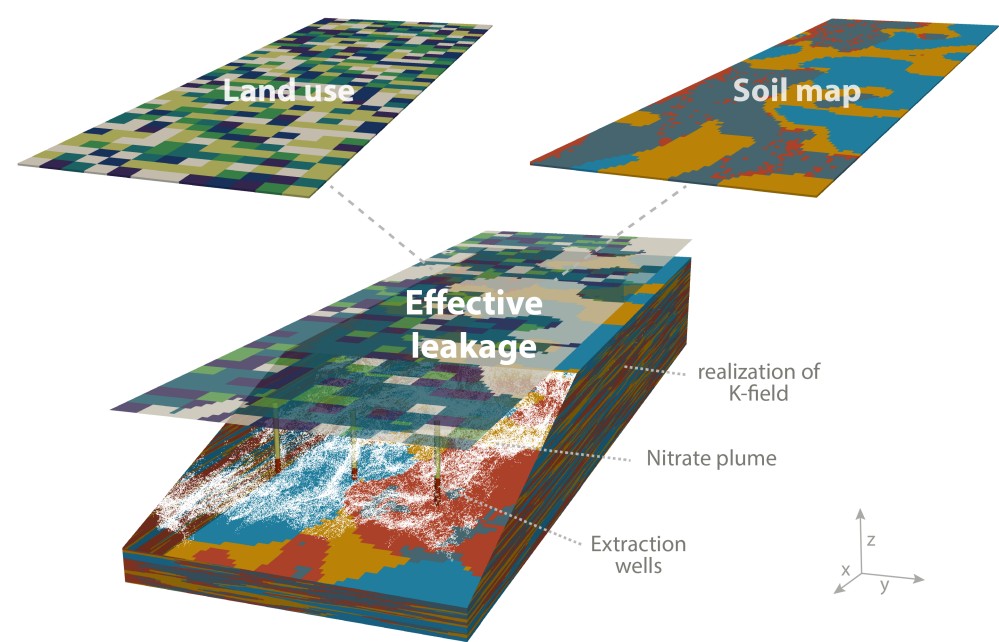

**Figure 1.** Illustration of the methodology used in the study, considering a representative quadri-lateral subsystem of a highly heterogeneous alluvial aquifer system, 19.2 x 6 x 0.25 km$^3$ (12 miles x 3.75 miles x 820 feet). A land use map is randomly generated (top-left). Each color represents a different crop. A soil map (top-right) is extracted from the top layer of each geostatistical realization of the hydraulic conductivity ($K$). For each combination soil-type/crop, effective leakage rates of water and nitrate are numerically estimated. White particles represent a snapshot of the NPS pollution (particles) eighty years after a single year of contaminant loading. In each simulation, three extraction wells are explicitly represented at downstream location of the domain. The lower part of each well (in red) represent its screen from where water is extracted and from where mass arrival is recorded. Other wells are implicitly represented by the flux into the lower boundary of the domain.

Flow and transport processes across each are solved using a specified (fixed), mappable land use representation. To assess the
spatial variability of NPS management metrics across an ensemble of well locations in a groundwater basin, the equiproba-
ble realizations of the aquifer system represent the variety of locations across a basin with geostatistically similar geological
features. In this case, land-use is simulated as a random process.

### 2.1.3 Aquifer

Spatial variability in the aquifer hydraulic conductivity ($K$) is represented using the transition probability/Markov chain method
(Carle, 1999) for generating random realizations of the hydrofacies field (Carle and Fogg, 1996, 1998). Here, we consider 4
hydrofacies: gravel, sand, muddy sand and mud. The geostatistical model requires the characterization of the proportion,
mean length and hydraulic conductivity of each facies, and of the facies-to-facies transition probability rates. We set these
parameters to be representative of Central Valley aquifer conditions (Weissmann and Fogg, 1999a, b) (Table 1 and 2). A total





**Table 2.** Mean length (diagonal values) of each categories (g: gravel; s: sand; ms: muddy sand; m: mud) and embedded transition probability (non-diagonal values) in the longitudinal (x), the horizontal transverse (y), and the vertical transverse (z) directions. Matrices reads as transition probability from the row facies to the column facies. The background category is designated by the letter $b$.

| (x) | g | s | ms | m |
|---|---|---|---|---|
| g | $\bar{L}_{g,x} = 800.0m$ | 0.7 | 0.15 | b |
| s | 0.7 | $\bar{L}_{s,x} = 1500.0m$ | 0.28 | b |
| ms | 0.15 | 0.28 | $\bar{L}_{ms,x} = 1000.0m$ | b |
| m | b | b | b | b |

| (y) | g | s | ms | m |
|---|---|---|---|---|
| g | $\bar{L}_{g,y} = 500.0m$ | 0.7 | 0.15 | b |
| s | 0.7 | $\bar{L}_{s,y} = 850.0m$ | 0.28 | b |
| ms | 0.15 | 0.28 | $\bar{L}_{ms,y} = 900.0m$ | b |
| m | b | b | b | b |

| (z) | g | s | ms | m |
|---|---|---|---|---|
| g | $\bar{L}_{g,z} = 2.0m$ | 0.7 | 0.15 | b |
| s | 0.7 | $\bar{L}_{s,z} = 3.5m$ | 0.28 | b |
| ms | 0.15 | 0.28 | $\bar{L}_{ms,z} = 2.0m$ | b |
| m | b | b | b | b |

of 50 realizations of the $K$-field were generated. An example of $K$-field can be observed in Figure 1. The histograms of the

mean and the variance of the logarithm of $K$ are shown in Figure SM3. Fifty realizations were sufficient to converge the lower statistical moments of $K$ and of the resulting mean velocities (Figure SM7).

#### 2.1.4    Soil map

The top layer of each $K$-field realization is here considered to represent the (unmapped) spatial variability of the soil type. Thus, a soil map, displaying the spatial distribution of the 4 hydrofacies at the land surface, is geostatistically consistent with

each realization of the aquifer $K$-field underlying the soil horizon.

#### 2.1.5    Land-use

The landscape of the simulated sub-basin is considered to be exclusively occupied by agricultural activities. Six different crop types are randomly distributed over a domain of 19200.0 m × 6000.0 m. The crops are: almond, citrus, corn, cotton, grain and grapes. All fields are of the same rectangular dimension, 360 m × 300 m (11 ha, 27 acres). The spatial distribution of crop

types is generated randomly and fulfills the following proportions of the 6 crop types: 24% of Almond, 24% of Citrus, 18% of Corn, 12% of Cotton, 12% of Grain, 10% of Grapes (Table 3). Crop types and proportion are representative of what may be encountered in the southeastern Central Valley (Harter et al., 2012).





### 2.1.6 Estimation of recharge and contaminant leakage

Numerical simulations were conducted to simulate the vadose zone flow and transport processes across all possible crop and
soil type combinations. Here, the gravel category in a soil layer was assumed to represent the same sandy soil as the sand
category. Hence, a total of 18 vadose zone profiles represent all possible combinations of the 6 different land types (crops)
and 3 different hydraulic soil profiles (sand, muddy sand, and mud). Simulations provide time-varying recharge and pollutant
leakage rates for each of the 18 possible land-use and soil combinations at the water table of the respective underlying aquifer
system. The time-series of the 18 simulations are computed *a priori* and then applied to define the water table boundary
conditions of the groundwater flow and transport simulations.

**Governing equations**

One dimensional water flow in soils is described by the Richards equation (Diamantopoulos and Durner, 2012):

$$\frac{\partial \theta}{\partial t} = \frac{\partial}{\partial z}\left[K'(\psi)\left(\frac{\partial}{\partial z}+1\right)\right] - S, \tag{1}$$

where $\theta$ [-] is the volumetric water content, $t$ [d] is time, $z$ [m] is the vertical spatial coordinate, positive upward, $\psi$ [m] is the
pressure head, $K'(\psi)$ [m$^2$d$^{-1}$] is the saturated/unsaturated hydraulic conductivity as a function of $\psi$ and $S$ is the sink term,
representing root water uptake [d$^{-1}$]. The water retention curve $\theta(\psi)$ and the hydraulic conductivity curve are required. The
two function are described by the van Genuchten-Mualem (van Genuchten, 1980) model:

$$\theta(\psi) = \begin{cases} \theta_r + (\theta_s - \theta_r) \times (1+ \mid \alpha\psi \mid^n)^{-m} & \psi < 0 \\ \theta_s & \psi \geq 0 \end{cases} \tag{2}$$

$$S_e = \frac{\theta(\psi) - \theta_r}{\theta_s - \theta_r} \tag{3}$$

$$K'(S_e) = K_s \times S_e^l \times \left[1 - \left(1 - S_e^{1/m}\right)^m\right]^2 \tag{4}$$

where $\theta_s$ and $\theta_r$ [-] are the saturated and residual water contents, respectively, $\alpha$ [m$^{-1}$], n [-], m [-], and l [-] are shape
parameters, $m = 1 - 1/n, n > 1$, and $S_e$ [-] is the effective saturation.

Solute transport for a conservative tracer is described using standard advection-dispersion equation of the form:

$$\frac{\partial \theta c}{\partial t} = \frac{\partial}{\partial z}\left(\theta D \frac{\partial c}{\partial z}\right) - \frac{\partial q c}{\partial z} - S \times c \tag{5}$$

where $c$ [g m$^{-3}$] is the concentration of the solute in the liquid phase, $D$ [m$^2$d$^{-1}$] is the dispersion coefficient, $q$ is the
volumetric water flux density (m d$^{-1}$) evaluated with the flow equation and $S \times c$ [g m$^{-3}$d$^{-1}$] is the root nutrient uptake for
the case of passive uptake. By focusing on hydrodynamic dispersion, $D$ is defined as

$$D = \lambda q/\theta \tag{6}$$

where $\lambda$ is the dispersivity [m].





**Table 3.** Area distribution, fertilization application and root zone depth for the 5 crops considered in this study.

| Crop | Area[a] (%) | Fertilization application[b] [kg/ha/year] | Root zone [cm] |
|---|---|---|---|
| Almond | 24.0 | 246.0 | 137.0 |
| Citrus | 24.0 | 157.0 | 107.0 |
| Corn | 18.0 | 239.0 | 91.0 |
| Cotton | 12.0 | 195.0 | 122.0 |
| Grain | 12.0 | 198.0 | 91.0 |
| Grapes | 10.0 | 39.0 | 91.0 |

[a] from Harter et al. (2012)

[b] from United States Department of Agriculture (1997)

**Parametrization of Hydrus 1D**

For the numerical solution of equations 1 and 5, we used the Hydrus 1D software (Šimunek et al., 2016). Root water uptake for the six different crops was simulated by assuming a macroscopic root water uptake approach (Feddes et al., 1978). The parameters for equations 2 and 5 were estimated by using Rosetta pedotransfer function (Scaap et al., 2004) and are shown in Table 4. For each soil horizon, dispersivity values were calculated by using the pedotransfer function of Perfect et al. (2002).

The simulation time was 21 years, from January 1, 1995 until December 31, 2015. Of the 21 years, the first 11 were used as a warm-up period and the remaining 10 were used to represent temporally variable boundary conditions at the top of the groundwater system. For an initial condition of equations 1 and 5, we assumed a uniform distribution of the pressure head and a solute free profile, respectively. The upper boundary condition for the flow problem accounts for precipitation, irrigation, and crop evapotranspiration. Daily reference (grass) evapotranspiration ($ET_0$) and precipitation ($P$) from the Stratford Meteorological station (California Irrigation Management Information System (CIMIS)) are used to represent southeastern Central Valley climate conditions. For each crop, $ET_0$ values were converted to potential crop evapotranspiration ($ET_c$) by using the single crop coefficient method (Allen et al., 1998). Daily time series of boundary conditions are used in Hydrus 1D. Based on calculated $ET_c$ and $P$ values, we created an irrigation schedule for each combination of crop-soil type, using the so-called evapotranspiration method (Allen et al., 1998). Irrigation was assumed to take place only during the crop period and not through the winter period (Figure 2). For all crop-soil combinations, we assume three fertilization events per year with the total amount of fertilizer application given in Table 3.

**Preparing Water Table Boundary Conditions from Unsatured Zone Simulation Results**

Simulations led to an estimation of the temporal evolution of water and nitrate leakage rate at the bottom of the 1D profile (Figure SM1 and SM2) for each crop - soil type combination, at daily time-steps. For the sake of simplicity, our groundwater simulation are conducted for steady state flow (but transient transport) conditions. Following Bastani and Harter (submitted), we homogenize both, recharge and pollutant flux in time and compute average, effective recharge and nitrate leakage rates over





**Table 4.** Horizon depth, soil hydraulic properties and dispersity of each horizon, for the three different soil profiles assumed in this study.

| Soil type | Horizons [cm] | $\theta_r$ [cm$^3$cm$^{-3}$] | $\theta_s$ [cm$^3$cm$^{-3}$] | $\alpha$ [cm$^{-1}$] | n [−] | $K_s$ [cm d$^{-1}$] | $l$ [−] | $\lambda$ [cm] |
|---|---|---|---|---|---|---|---|---|
|  | 0-20 | 0.036 | 0.389 | 0.033 | 1.41 | 20.30 | -1.03 | 2.0 |
|  | 20-56 | 0.035 | 0.389 | 0.036 | 1.43 | 23.44 | -1.06 | 3.6 |
| S1 | 56-86 | 0.038 | 0.388 | 0.034 | 1.42 | 20.39 | -1.07 | 3.0 |
|  | 86-147 | 0.035 | 0.389 | 0.038 | 1.44 | 25.49 | -1.06 | 6.1 |
|  | 147-1000 | 0.033 | 0.391 | 0.040 | 1.47 | 29.72 | -1.03 | 3.0 |
|  | 0-30 | 0.039 | 0.376 | 0.057 | 1.66 | 50.17 | -1.08 | 3.0 |
| S2 | 30-107 | 0.048 | 0.370 | 0.055 | 1.86 | 46.55 | -1.07 | 7.7 |
|  | 107-1000 | 0.032 | 0.343 | 0.056 | 1.66 | 46.66 | -1.07 | 4.3 |
|  | 0-23 | 0.091 | 0.4988 | 0.022 | 1.19 | 7.86 | -3.01 | 2.3 |
| S3 | 23-94 | 0.084 | 0.4978 | 0.014 | 1.22 | 0.09 | -1.58 | 7.1 |
|  | 94-1000 | 0.094 | 0.4740 | 0.018 | 1.27 | 4.28 | -1.52 | 5.6 |

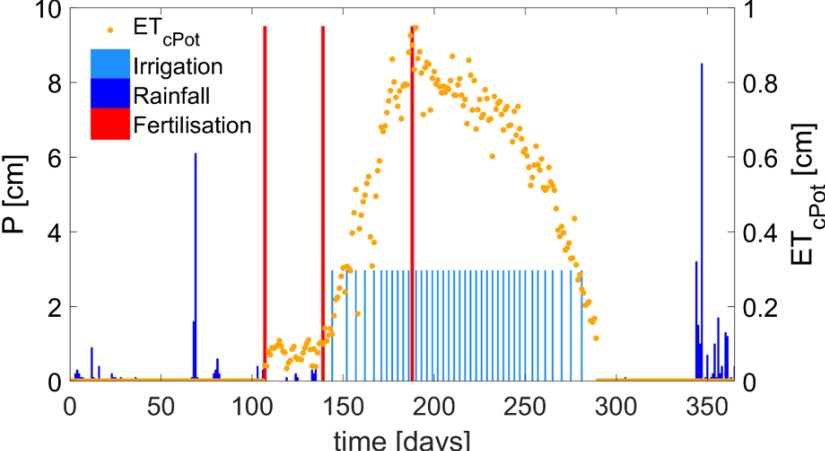

**Figure 2.** Illustration example of daily values of potential crop evapotranspiration (orange points) used for the description of the upper boundary condition. The blue bars represent rainfall events and the light blue bars irrigation events. The red bar defines fertilization application for an amount equal to 65 Kg/ha (195 Kg/ha/year). Crop is cotton.





**Table 5.** Recharge rate and nitrate mass flux applied for each crop - soil type combination

| $r$ [m/m$^2$/d] | almond | citrus | corn | cotton | grain | grapes |
|---|---|---|---|---|---|---|
| S1 | $2.3\times10^{-3}$ | $1.7\times10^{-3}$ | $1.5\times10^{-3}$ | $1.7\times10^{-3}$ | $4.1\times10^{-4}$ | $1.8\times10^{-3}$ |
| S2 | $1.7\times10^{-3}$ | $1.4\times10^{-3}$ | $1.3\times10^{-3}$ | $1.4\times10^{-3}$ | $5.0\times10^{-4}$ | $1.4\times10^{-3}$ |
| S3 | $2.9\times10^{-5}$ | $2.3\times10^{-4}$ | $2.6\times10^{-4}$ | $2.0\times10^{-4}$ | $2.0\times10^{-5}$ | $1.5\times10^{-4}$ |

| $m_f$ [g/m$^2$/d] | almond | citrus | corn | cotton | grain | grapes |
|---|---|---|---|---|---|---|
| S1 | $2.8\times10^{-2}$ | $2.5\times10^{-2}$ | $2.0\times10^{-2}$ | $2.1\times10^{-2}$ | $1.4\times10^{-2}$ | $4.6\times10^{-3}$ |
| S2 | $2.8\times10^{-2}$ | $2.1\times10^{-2}$ | $1.5\times10^{-2}$ | $1.4\times10^{-2}$ | $1.8\times10^{-2}$ | $4.3\times10^{-3}$ |
| S3 | $0.0$ | $1.9\times10^{-10}$ | $3.3\times10^{-9}$ | $0.0$ | $0.0$ | $0.0$ |

a 10 year time series (illustration in Figure SM1 and SM2 and average values in Table 5). For each of the 50 $K$-field (and, therefore, soil map) realization, the temporally homogenized results for each crop-soil combination are then used to describe the spatial distribution of the effective recharge ($r(x,y)$) and nitrate mass flux ($m_0(x,y)$). Histograms of the mean and the variance of recharge rate ($r$) and initial concentrations ($c_0$) are shows in Figure SM4 and SM5, respectively.

## 2.2 Groundwater flow and transport

### 2.2.1 Flow

Groundwater flow and nitrate transport are numerically simulated. We consider a 3-dimensional aquifer with a length ($L_x$) of 19200.0 m, a width ($L_y$) of 6000.0 m, and a depth ($L_z$) of 250.0 m (Table 6). Average steady-state groundwater flow conditions are governed by (Rushton and Redshaw, 1979):

$$\frac{\partial}{\partial x}\left(K_{xx}\frac{\partial h}{\partial x}\right) + \frac{\partial}{\partial y}\left(K_{yy}\frac{\partial h}{\partial y}\right) + \frac{\partial}{\partial z}\left(K_{zz}\frac{\partial h}{\partial z}\right) + Q'_s = 0, \tag{7}$$

where $h$ (m) is the hydraulic head, and $Q'_s$ is a volumetric flux per unit volume representing sources and sinks of water. Groundwater fluxes are simulated by solving numerically the Darcy's law:

$$\mathbf{q}(\mathbf{x}) = -K(\mathbf{x})\nabla h(\mathbf{x}), \tag{8}$$

where $q$ (m d$^{-1}$) is the specific discharge in the 3 dimensions x={x,y,z}. The longitudinal flow is defined by a regional gradient of $1.0\times10^{-3}$. The vertical flow is impacted by recharge and by a downward flux from the bottom of the domain. The spatially distributed fixed flux boundary condition across the bottom of the domain represents water flux to pumping wells in the deeper part of the aquifer and the effect of implied, non-simulated groundwater extraction by wells distributed throughout the lower part of the simulated aquifer sub-basin. Domestic wells are not considered to significantly affect flow and transport processes and are not simulated. Recharge is considered spatially variable to account for realistic spatial distribution of crop and soil types (see Section 2.1.6). Histograms of the mean and variance of the recharge rates applied in the 50 realizations of heterogeneous cases are shown in Figure SM4.





**Table 6.** Domain discretization and physical parameters used in all simulations

| Parameter | Value |
|---|---|
| *Domain discretization:* | |
| Number of cells, $n_x \times n_y \times n_z$ | $120 \times 60 \times 625$ |
| Cell dimension, $\Delta_x \times \Delta_y \times \Delta_z$ [m×m×m] | $160.0 \times 100.0 \times 0.4$ |
| Domain length, $L_x \times L_y \times L_z$ [m×m×m] | $19200.0 \times 6000.0 \times 250.0$ |
| *Flow and transport problem:* | |
| Porosity, $\phi$ [-] | 0.3 |
| Average longitudinal hydraulic gradient, $i_x$ [-] | $1 \times 10^{-3}$ |
| Extraction rate, $Q_{out}$ [m³d⁻¹] | $3 \times 10^3$ |

Three extraction wells are located near the downstream edge of the domain. The extraction rate, $Q_{out}$, is set to 3000 m³d⁻¹ (551 gpm), corresponding to an actual irrigation pumping rate of 6000 m³d⁻¹ (1102 gpm) or 9000 m³d⁻¹ (1653 gpm) during

a six month or four month annual irrigation season, respectively. The length of the well screen is location and realization dependent, depending on the vertical distribution of highly conductive material (gravel, sand) (Appendix A). The total outflow (downward flux at the domain bottom plus extraction at the three wells) is set to be equal to the inflow of water by recharge. The resulting water flow system representation is solved using MODFLOW (Harbaugh et al., 2000) for each realization of the $K$-field and the upper boundary.

**2.2.2   Transport**

Nitrate transport is modeled using the advection-dispersion equation (ADE) given by:

$$\phi \frac{\partial c}{\partial t} = \nabla \cdot (\phi \mathbf{D} \nabla c) + \nabla \cdot (\mathbf{q} c) \tag{9}$$

where $c$ is the solute concentration, $\mathbf{D}$ is the 3-dimensional dispersion tensor, and $\phi$ is the porosity. The ADE is solved by the random walk particle tracking (RWPT) method implemented in the Fortran code RW3D (Fernàndez-Garcia et al., 2005; Henri

and Fernàndez-Garcia, 2014, 2015). RWPT solves the ADE by moving a large number of particles in successive jumps given by: (e.g., Salamon et al., 2006).

$$\mathbf{x}_p(t + \Delta t) = \mathbf{x}_p(t) + \Delta t [\mathbf{v}(\mathbf{x}_p(t)) + \nabla \cdot \mathbf{D}(\mathbf{x}_p(t))] + \sqrt{\mathbf{D}(\mathbf{x}_p(t)) \Delta t} \cdot \xi(t), \tag{10}$$

where $\mathbf{x}_p$ is the particle position, $\mathbf{v}$ is the velocity vector, and $\xi$ is a normally distributed random variable with zero mean and unit variance.

The detailed discretization of the velocity field described above is capturing the most relevant characteristics affecting the macro-dispersive transport behavior (LaBolle, 1999; LaBolle and Fogg, 2001; Weissmann et al., 2002; Henri and Harter, 2019). Therefore, effects of grid-scale dispersion are assumed to be negligible. This assumption, which potentially impacts NPS management metrics, is further evaluated in Appendix B.





Nitrate transport originating from the water table is simulated using an instantaneous injection of 500,000 particles over
the entire top of the domain. Particle transport is tracked using the RWPT algorithm (Eq. 10) for a simulation time of 350
years. In simulations with spatially variable initial contaminant mass loading, the local density of particles reproduces the
local initial concentration ($c_0(x,y)$) in recharge water at the groundwater table. Histograms of the mean and variance of the
initial concentration over the 50 realizations are shown in Figure SM5. Following the superposition principle, the cumulative
mass arrival at wells resulting from an instantaneous injection of mass $m_0$ can be interpreted as $\dot{m}(t)$, the simulated mass
flux at wells resulting from a continuous and temporally constant release of mass $m_0$. Well concentrations are computed as
$c_w(t) = \dot{m}(t)/Q_{out}$. Flow and transport are simulated for each realization of the $K$-field and water table boundary condition.

## 2.3 Nonpoint source pollution management metrics

### 2.3.1 Pollutant travel times

Three relevant nonpoint source (NPS) pollution management metrics are considered to measure the stochastic simulation
outcomes: the probability distribution of pollutant travel times to wells, the probability distribution of pollutant concentration
in wells, and the probability distribution of source locations. The probability distribution of travel times is obtained by recording
travel times of particles to the compliance area, that is, the screen of the extraction well, for each particle. We obtain normalized
travel times $t_i$ by computing the time required to observe a specified fraction $i$ of the total mass that reaches a well over the total
350 year simulation period. For instance, t5 represents the travel time from the water table to the well for the fifth percentile
of the total mass reaching the well in 350 years. Following a stochastic approach, probability density functions (*pdf*s) of travel
times $t_i$ are obtained by determining the histogram of $t_i$ in 150 simulated wells (50 realizations, each with 3 wells). Travel time
*pdf*s can represent a useful tool to assess both the expected time of solute arrival at the compliance area and the propagation of
uncertainty from the hydraulic conductivity field to pollutant transport (e.g. Dagan and Nguyen, 1989; Cvetkovic et al., 1992;
de Barros and Rubin, 2008; Henri et al., 2016).

### 2.3.2 Pollutant concentrations in wells (breakthrough curves)

The assessment of potential contaminant levels in extraction wells represents a key step in NPS pollution management. Under
uncertain flow conditions, managers would benefit from knowing the probability to exceed a threshold concentration such as
the maximum contaminant level (MCL) in a given well or in a series of wells. For each of the 150 simulated wells, breakthrough
curves $c(t)$ are obtained. Their probability distribution $P_i(c,t)$ is obtained as a sample distribution of $c(t)$, where $P_i(c,t)$ is the
probability for $i\%$ of wells to exceed the contaminant level $c$ at time $t$.

### 2.3.3 Capture zones

NPS pollution management may also require the assessment of the effective source area, i.e., the capture zone or contributing
area of the pollution observed in a production well. The spatial variability of hydraulic properties leads to uncertainty about and
spatial variability of the source area (e.g. Varljen and Shafer, 1991; Franzetti and Guadagnini, 1996; Riva et al., 1999; Stauffer





et al., 2002). In the stochastic framework, the capture zone is assessed by defining the spatial distribution of the probability that a contaminant leaking from the NPS will reach a well ($P_w$), i.e.

$$P_w(\mathbf{x}_{\text{NPS}}) = \text{Prob}(\mathbf{x}_p(t \in [0, t_{end}]) = \mathbf{x}_w \mid \mathbf{x}_p(t = 0) = \mathbf{x}_{\text{NPS}}), \tag{11}$$

where $\mathbf{x}_p$ is the 3-dimensional location (in the Cartesian coordinate system given by $\mathbf{x} = (x, y, z)$) of a portion of the plume (represented by a particle in this study), $\mathbf{x}_w$ is a location shared with a well screen, and $\mathbf{x}_{\text{NPS}}$ is a given location of the NPS. The spatial extension of non-zero probabilities forms then a probabilistic capture zone. The time required for a contaminant leaving a given location of the contributing area to reach the extraction well (or so-called time-related capture zone) is also stochastically analyzed.

## 3 Upscaling and test cases

### 3.1 Homogenization of source terms

The NPS metrics from fully heterogeneous simulations are compared to the NPS metrics obtained from a range of upscaled, homogenized simulations that employ effective homogeneous properties rather than the original heterogenous distribution of the $K$, $r$, and $c_0$ terms. The source terms ($r(x,y)$, $c_0(x,y)$) are homogenized separately for each realization by spatial averaging to obtain $\langle r \rangle$ and $\langle c_0 \rangle$. Histograms of $\langle r \rangle$ and $\langle c_0 \rangle$ show significant variability of the homogenized source terms between realizations (Figures SM3 and SM4). Homogenized recharge rates and source concentrations range from 0.9 to 1.4 m d$^{-1}$ m$^{-2}$ and from 5.0 to 8.5 g m$^{-3}$, respectively. A number of different homogenized models are considered and compared against the reference case:

- a heterogeneous $r$ and heterogeneous $c_0$ (reference case);

- a heterogeneous $r$ and homogeneous $c_0$;

- a homogeneous $r$ and heterogeneous $c_0$;

- a homogeneous $r$ and homogeneous $c_0$;

### 3.2 Homogenization of the hydraulic conductivity and transport upscaling

To simulate flow and transport in an equivalent homogeneous, upscaled $K$ conditions, we estimate the effective longitudinal and transverse vertical hydraulic conductivity, $K_x^*$ and $K_z^*$, and dispersion, $\alpha_L^*$ and $\alpha_{TV}^*$. Effective parameters in the longitudinal direction ($K_x^*$ and $\alpha_L^*$) are determined from the first and second spatial moments of a plume resulting from an injection of mass in a vertical plane of width 3000.0 m and depth 50.0 m. The same approach is adopted to estimate the effective parameters in the transverse vertical direction ($K_z^*$ and $\alpha_{TV}^*$) by injecting particles in a horizontal plane covering the entire top of the domain. No extraction is considered in both cases in order to capture the natural behavior of the plume. For each realization of the $K$-field, the slope of the temporal evolution of the first spatial moment, i.e. the plume center of mass, is used to evaluate the





apparent velocities, $v_x^*$ and $v_z^*$. After estimation of the gradients from simulated head differentials in the $x$- and $z$-directions,

Darcy's law is applied to evaluate effective hydraulic conductivities $K_x^*$ and $K_z^*$ (Eq. 8). Effective dispersion values ($\alpha_L^*$ and $\alpha_{TV}^*$) are similarly obtained by analyzing the slope of the normalized second spatial moment of particle plume. The importance of representing upscaled dispersion is independently tested. Histograms of resulting of upscaled $K_x^*$, $K_z^*$, $\alpha_L^*$, and $\alpha_{TV}^*$ values as well as the satisfactorily convergence of the mean of the apparent parameters after 50 realizations are shown in Figures SM6 and SM7.

Furthermore, the cumulative implication of homogenization in aquifer properties and in the source terms $c_0$ and $r$ is tested. The series of scenarios considered are:

- a heterogeneous $K$, a heterogeneous $r$ and a heterogeneous $c_0$ (reference case);

- an upscaled $K$, $r$ and $c_0$, considering advection only;

- an upscaled $K$, a heterogeneous $r$ and $c_0$, considering advection only;

- an upscaled $K$, a heterogeneous $r$ and $c_0$, considering advection and upscaled dispersion.

## 4 Results and Discussion: Homogenization of source terms

The effect of conceptually simplifying recharge, contaminant input concentration, and aquifer heterogeneity on the stochastic description of travel times, well concentrations, and capture zones is here illustrated specifically for the case of quantifying uncertainty about these NPS pollution management metrics at a particular well surrounded by a spatially distributed, but fixed
(known) distribution of land use across all realizations (scenario "LU 1"). Alternatively, the effects of homogenization on the analysis of spatial variability across an ensemble of wells in a groundwater basin, where land use is different in each realization (scenario "LU 50"), is further discussed in Section 6.

### 4.1 Travel time

We analyze the probability distribution of travel time for the 5%ile ($t_5$), 50%ile ($t_{50}$) and 95%ile ($t_{95}$) mass reaching a well
within the 350 year simulation period (Figure 3). These metrics characterize the temporal variability of the early, median, and late mass travel time from of a one-year pollutant (e.g., nitrate) input to the aquifer system. For all simulations, early mass travel times are within a range of 10 to 100 years with an expected value (highest probability) of 50 years (Figure 3a). Late mass travel times are likely to be in the range of 50 to 300 years (Figure 3c), with an expected travel time of about 120 years. These results are roughly consistent with the estimation of groundwater age distribution made by Weissmann et al. (2002) in
the San Joaquin Valley from detailed modeling and from chlorofluorocarbon (CFC) age data (mean groundwater ages of 10 to 50 years in twice to three times shallower wells than the ones simulated in this study).

The homogenization of recharge spatial variability directly affects the flow field in the uppermost part of the aquifer. While the effect is larger than the homogenization of the concentration (see next paragraph), it also has no significant impact on travel



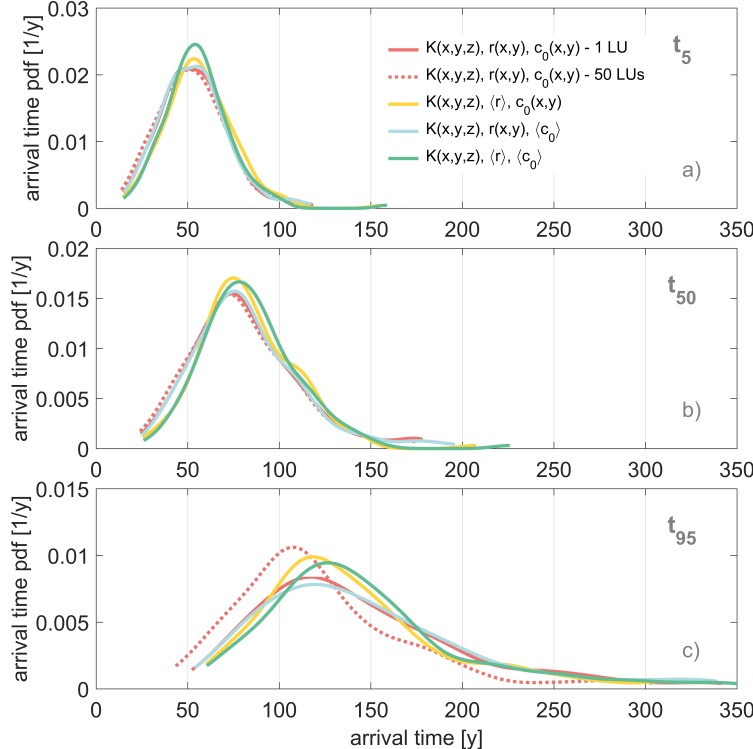

**Figure 3.** Probability density function of the time required for 5% (a), 50% (b), and 95% (c) of the total recorded mass to reach a well considering a heterogeneous $r$ and heterogeneous $c_0$ (reference case, red line); a homogeneous $r$ and heterogeneous $c_0$ (yellow line); a heterogeneous $r$ and homogeneous $c_0$ (light blue line); a homogeneous $r$ and homogeneous $c_0$ (green line). Plain lines refers to the consideration of an identical land-use map for all realizations. For comparison, the red dash lines shows outputs from simulations with a realization-dependent land-use maps.

time $pdf$s (Figure 3): The distributions are slightly less spread out over the time axis, with slightly higher and earlier mode

(peak of the $pdf$) and lower probabilities in the tails of the pdf (probability differences at all times $< 5\%$), especially of the late travel time (probability differences at all times around 10%).

The homogenization of initial concentrations has no physical impact on travel times since it does not affect the velocity field in the groundwater system. However, the difference in input concentration changes the distribution of the initial mass across the water table. Hence, there are small but discernible differences in the travel time $pdf$s of variable and homogeneous $c$ both,

in the case of spatially variable $r$ (red and blue lines in 3) and in the case of homogeneous $r$ (yellow and green lines in 3).

Importantly, the homogenized representation of $r$ and $c$ has nearly no effect on the time span between early and late arrival times at the well screen (the contrast in the position of the travel time $pdf$s for $t_5$ and $t_{95}$), which represents the age difference





between the youngest and the oldest water captured by the well screen and then mixed during the pumping process (Weissmann et al., 2002; Koh et al., 2018; Henri and Harter, 2019).

## 4.2 Stochastic capture zone

The stochastic capture zone is the area characterized by $P_w(x, y) > 0$. Simulation results for the fully stochastic representation of source heterogeneity show that the stochastic capture zone covers an area of about 8000 by 4000 m (about 30% of the simulated domain, containing approximately 300 individual fields), while the zone from where mass is the most likely to reach a well (critical zone, $P_w(x, y) > 0.5$) is more spatially focused (about 3% of the simulated domain, the size of about 30 individual fields, see Figure 4).

As explained above, homogenizing the input concentration does not affect the velocity field and transport processes, and only slightly reduces $P_w$ values of the critical zone (Figure 4b). On the other hand, not accounting for spatial variability in recharge leads to an overestimation of $P_w$ values inside the critical zone (Figure 4c). The same observation is made when both $r$ and $c_0$ are considered homogeneous (Figure 4d). The location of the critical zone, being controlled by regional flow conditions and well characteristics (extraction rate, depth and length of the screen), is not impacted by the spatial description of source terms. Spatial variability in the recharge is responsible for somewhat more uncertainty in the exact delineation of the capture zone along its margins than what is captured by the homogenization of $r$.

Recharge rates, if considered heterogeneous, are by design correlated to the hydraulic conductivity. Highly conductive material are associated with high recharge rates, which may increase the channeling effect through preferential paths. Accounting for spatial variability in the recharge rate will, therefore, exacerbate the impact of the heterogeneity in the $K$-field, especially near the water table (where recharge is applied), thus increasing the uncertainty about delineating the source area.

Just as travel time $pdf$s are little affected (Figure 3), the overall location of the stochastic capture zone is approximated quite well with the homogenized parametrization of concentration and recharge. Consequently, the average travel times required for a particle leaving a given location of the NPS to reach a well also are not dramatically impacted by the spatial representation of the two source terms (Figure 5). The average flow condition is common to all simulations. Since the top of well screens is 100 m deep, the solute transport from the source to the compliance areas occurs mostly at depths far away from the spatially variable top boundary condition, where local change of flow condition at the surface does not impact significantly groundwater fluxes.

## 4.3 Well NPS pollution concentration

A common characteristic of NPS pollution different from many point source cases is the temporal continuity and consistency in NPS inputs. For example, significant nitrate loading to groundwater began with the introduction of commercial fertilizer just before World War II and has continued since then (Rockstrom et al., 2009; Harter et al., 2017). The long-term consequence of such continuous NPS loading, year-after-year, can be obtained from our simulations by superpositioning breakthrough curves obtained for NPS output in a single year at t = 0. If we neglect long-term trends or year-to-year variations in NPS and assume a constant input of nitrate to the water table, then the stochastic breakthrough curve at the well screen is simply obtained by



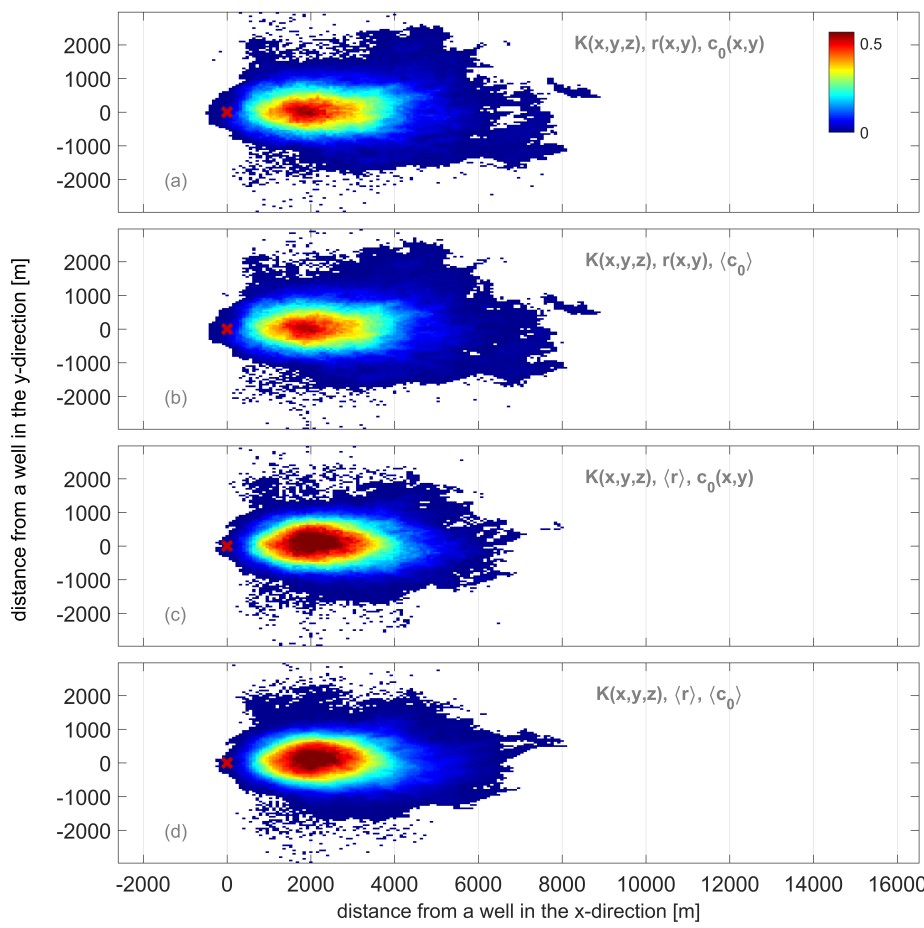

**Figure 4.** Probability of a particle leaving a given grid-cell to reach a well accounting for a heterogeneous $r$ and heterogeneous $c_0$ (reference case, a); a heterogeneous $r$ and homogeneous $c_0$ (b); a homogeneous $r$ and heterogeneous $c_0$ (c); a homogeneous $r$ and homogeneous $c_0$ (d); a $lnK$-weighted $r$ and $lnK$-weighted $c_0$ (e).



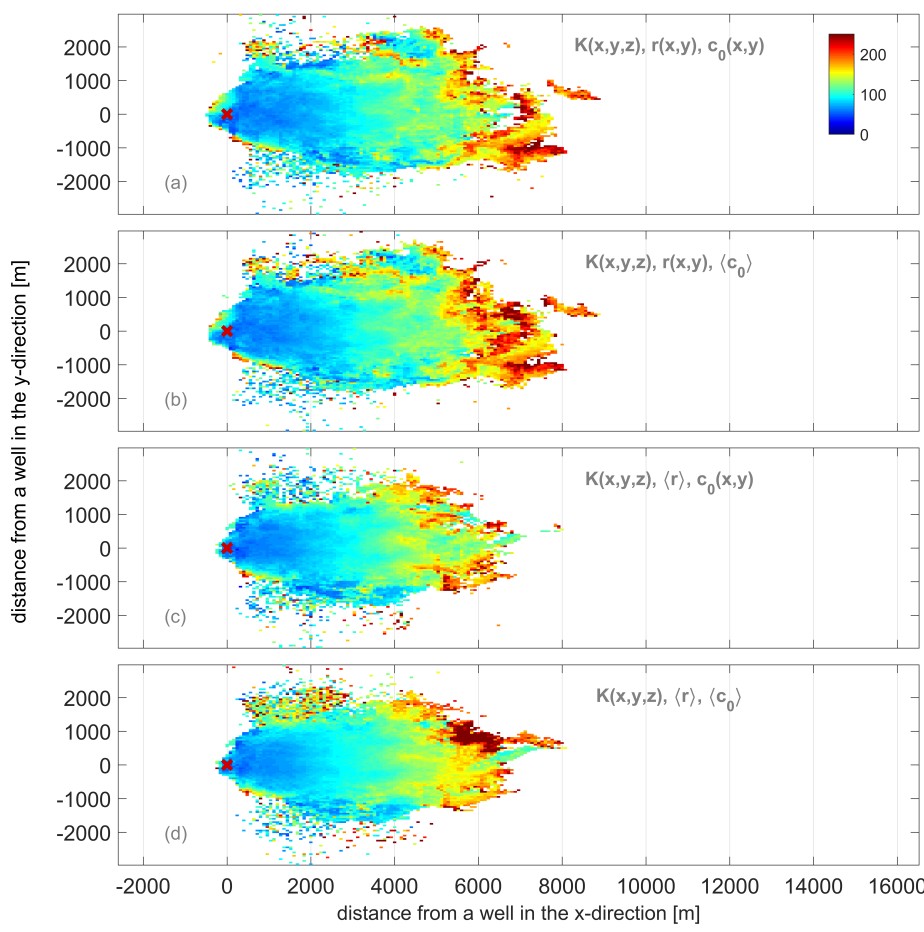

**Figure 5.** Expected travel time [years] of a particle leaving a given grid-cell and reaching a well for a heterogeneous $r$ and heterogeneous $c_0$ (reference case, a); a heterogeneous $r$ and homogeneous $c_0$ (b); a homogeneous $r$ and heterogeneous $c_0$ (c); a homogeneous $r$ and homogeneous $c_0$ (d); a $lnK$-weighted $r$ and $lnK$-weighted $c_0$ (e).





computing the cumulative distribution function (CDF) of the concentration $pdf$ (Henri and Harter, 2019) (Figure 6). The CDF plots provide a measure of the expected time at which a given threshold contaminant level (in the $x$-axis) such as the MCL (10 mg/L for nitrate as nitrogen in the U.S.) will be exceeded with a probability of 90% (P90, left), 50% (P50, center), or 10% (P10, right). In a regional context, these graphs can be interpreted as the time ($x$-axis) after which at least 90%, 50%, and 10%

of all wells in the aquifer region exceed a concentration of interest ($y$-axis), respectively.

For the reference scenario (red curve in Figure 6), we observe that the concentration eventually exceeded by 90% of wells is about 4.5 mg N/L, after about 250 years (left graph in Figure 6). Half of the wells will have a concentration exceeding 11 mg N/L (again, after about 250 years). Also in at least half of wells, the onset of rising nitrate levels (to above background levels) will occur no later than 50 years after the start of nitrate loading, reaching levels corresponding to half of the MCL (5

mg N/L) after about 70 years, and reaching the MCL (10 mg N/L) no later than about 150 years (middle graph in Figure 6). The 10% most nitrate contaminated wells will show an onset of nitrate contamination no later than 30 years after the start of NPS pollution, exceed the MCL in less than 70 years, and reach concentrations exceeding 14.5 mg N/L no later than about 150 years (right graph in Figure 6).

For an individual well, the results indicate that there is a 10% chance for nitrate concentrations to start to rise before 30

years, a 50% chance to rise no later than 50 years, and a 90% chance to rise before 70 years. Similarly, results suggests that the MCL will be exceeded with 10% probability after 70 years and with 50% probability after 140 years.

These results are consistent with observations of nitrate concentrations in drinking water and irrigation wells in the San Joaquin Valley, the southern half of the Central Valley. In Merced, Stanislaus, Tulare, and Kings County, about 40% of domestic wells (with screen depths not exceeding 100 m) exceed the drinking water standard (Ransom et al., 2013), but only about 10%

of the large production wells in the southeastern San Joaquin Valley (the wells represented in this study) exceed the nitrate MCL (10 mg N/L) (Survey et al., 2012, 2013), approximately 70 years after the beginning of extensive fertilizer use in the region. We note that the time scale for these concentration increases is very sensitive to two aquifer parameters: the hydraulic conductivity and the average effective porosity. If the regional average $K$ was twice as large as assumed in our model, all times would be half as long. Similarly, if the average regional effective aquifer porosity was 20% larger, travel times would be 20%

shorter.

The homogenization of spatial variability in the recharge rate and in the source concentration, while of limited consequence to travel time estimates and to estimates of source area extent, has measurable implications for stochastic well concentration predictions, particularly at the lower margin: Homogenizing the recharge rate only leads to significantly (>40%) underestimating the maximum concentration exceeded by 90% of wells in the intermediate and long term. Homogenization leads to

somewhat (≈10%) overestimating the concentration exceeded by 10% of wells over the long term, but reproduces well the concentration exceeded by 50% of wells at all time (yellow vs red in Figure 6).

Homogenizing both, recharge and contaminant loading does not affect the predictions quite as much, and in the opposite direction: The (lower) concentrations exceeded by 90% of wells are overestimated and the (higher) concentrations exceeded by only 10% of wells are underestimated, while the concentrations exceed by 50% of wells are less than 5% different from the

fully stochastic prediction.





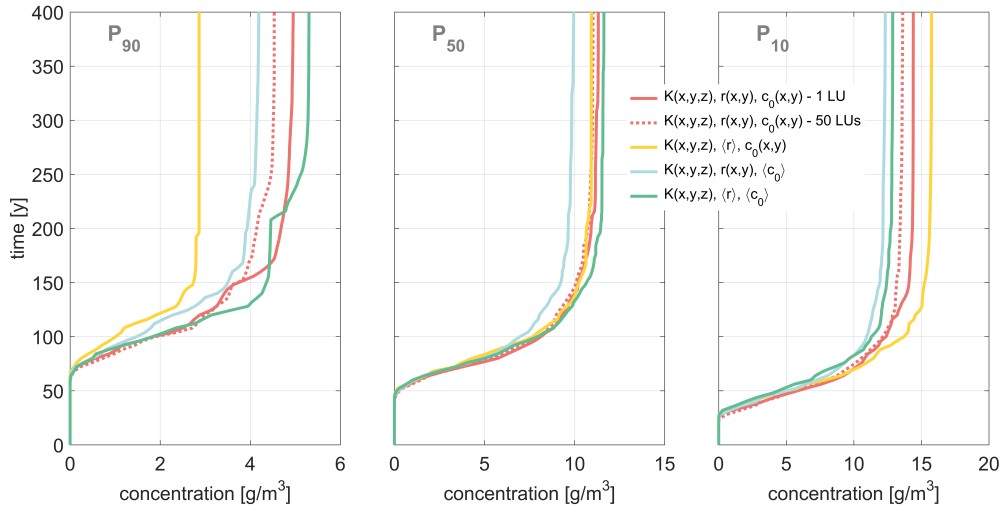

**Figure 6.** Time ($y$-axis) required for a well to exceed a given concentration ($x$-axis) with a probability of 90% (left), 50% (middle) and 10% (right) considering a heterogeneous $r$ and heterogeneous $c_0$ (reference case, red line); a homogeneous $r$ and heterogeneous $c_0$ (yellow line); a heterogeneous $r$ and homogeneous $c_0$ (light blue line); a homogeneous $r$ and homogeneous $c_0$ (green line). For comparison, the red dash lines shows outputs from simulations with a realization-dependent land-use maps.

Results are also sensitive to a homogenization of only the initial concentrations, which would underestimate all concentrations by about 10% (blue lines in Figure 6). Homogenizing only concentration also leads to an underprediction, by about 20%, of concentrations exceeded by either 90%, 50%, or 10% of well, relative to the fully stochastic land use treatment (green lines in Figure 6).

The results of the homogenization and the differences to treating land use in fully stochastic mode ("50 LUs") are driven directly by three factors: the distribution of land use, including the size of fields relative to the source area, and the distribution of recharge and nitrate leaching among different land uses. As shown in Section 4.2., the extent of the capture zone encompasses hundreds of fields, while the critical capture zone – the core contribution area – encompasses at least 30 fields. For field size much larger than those simulated here, or for a more spatially correlated distribution of crops among fields, homogenization

across all land uses in a basin may lead to larger errors due to the smaller number of land use "samples" intersected by the capture zone (Gibbons, 1994). Furthermore, unsaturated zone flow and transport simulations have led to highest contaminant leakage rates in areas of high recharge (almonds, citrus, see Table 5). Homogenizing mass leakage therefore decreases the amount of contaminant in high recharge areas and consequently globally underestimates well concentrations. Outcomes would be different if the highest concentration is associated with the lower recharge rate.

The examples shown here indicate that there may be significant errors in predicting future concentrations exceeded by 90% of wells and by 10% of wells, i.e., the distribution of exceedance probabilities among the ensemble of wells, whereas the concentration exceeded by half of the wells is characterized quite accurately under homogenized land use treatment. Overall,





the homogenization of recharge in particular leads to the largest potential errors of NPS pollution management metrics, less so
for predicting travel times and capture zone, but significantly so for predicting the distribution of exceedance concentrations

across an ensemble of wells.

## 5   Results and Discussion: Homogenization of K

### 5.1   Travel time

In a second step, the implications of upscaling aquifer heterogeneity on the stochastic description of travel times, capture zones
and well concentrations are assessed. Probability density functions of early, medium and late travel times are significantly

impacted by the full homogenization of the hydraulic conductivity field (Figure 7). A homogenization of both, aquifer and land
use random processes ($K$, $r$, $c_0$) drastically reduces the spread of all mass percentile travel times (yellow lines in Figure 7).
But the homogenized prediction of modes is quite accurate: The mode of the early mass and median mass travel times ($t_5$ and
$t_{50}$) are predicted with about 10% accuracy relative to the fully stochastic solution (Figure 7a and b). For the late mass travel
times, the mode in the homogenized prediction occurs later than for the case of a fully heterogeneous system (Figure 7c).

Aquifer heterogeneity generates a complex network of well-connected channels but also zones of near-stagnation, all of
which controls the spatio-temporal behavior of contaminant plumes across all scales. The effective solute path architecture
is, therefore, specific to the $K$-field realization and highly uncertain. In the stochastic solution, this generates a large range
of probable solute travel times (travel time $pdf$s with large variance) to the well screen that cannot be capture by simulating
transport in a homogenized $K$ architecture (Figure 7).

However, the global motion of the plume, characterized by its first spatial moment and the downward movement of the
first moment along the the depth interval of the well screen over time would be approximately similar for all realizations,
given the geostatistical parameters and regional gradients. Thus, accounting for upscaled advective motion only (obtained
from the estimation of the first spatial moment) preserves the large mixing in the well screen (Figure A1), but underestimates
the uncertainty on travel times arising from the macro-dispersive effects of heterogeneity. This is captured by the fact that

the modes of the early, median, and late mass arrivals are spread over similar time periods (45 years to 140 years), even the
prediction based on a completely homogenized representation of both, aquifer and land use processes captures a significant
fraction of the age distribution of mass arriving at the well screen. This is due to the significant mixing that occurs in the well
screen when the well is being pumped (Weissmann et al., 2002; Henri and Harter, 2019).

Similar results to those for a fully homogenized representation are found when only the $K$-field is homogenized, but land

use is represented with heterogeneous $r$ and $c_0$. The spread of each mass percentile travel time $pdf$ is slightly larger than in the
fully homogenized case, but is relatively far from capturing the full extent of the travel time $pdf$s for the fully heterogeneous
simulations (compare blue and red lines in Figure 7).

While the homogenization of $K$ removes the controlling process of the macro-dispersive pollutant behavior, the macro-
dispersive behavior can be approximated by including an upscaled, homogenized dispersion process (Eq. 9) into the simulation

(Eq. 10). Using both, homogenized $K$ and a representative, effective macro-dispersion much improves the accuracy of the





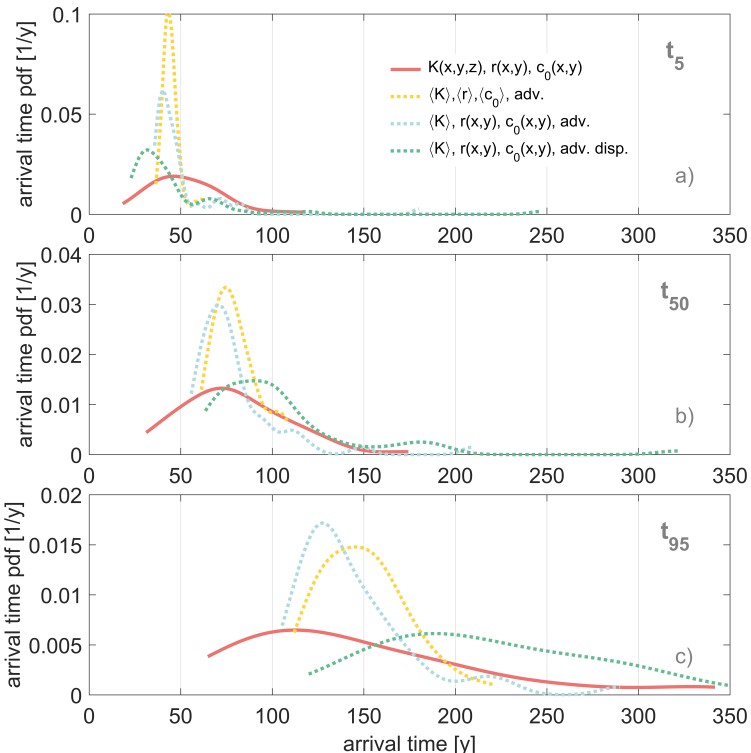

**Figure 7.** Probability density function of the time required for 5% (a), 50% (b), and 95% (c) of the total recorded mass to reach a well considering a heterogeneous $K$ field, heterogeneous $r$ and heterogeneous $c_0$ (reference case, red solid line); an upscaled $K$ values, averaged $r$ and averaged $c_0$ accounting for advection only (yellow dashed line); an upscaled $K$ values, heterogeneous $r$ and heterogeneous $c_0$ accounting for advection only (blue dashed line); an upscaled $K$ values, heterogeneous $r$ and heterogeneous $c_0$ accounting for advection and dispersion (green dashed line).

homogenized prediction and captures significant features of the fully stochastic prediction (green lines in Figure 7). Early mass travel times are slightly underestimated while median and especially late mass travel times are slightly overestimated.

Applying second spatial moments from heterogeneous simulations to estimate the macrodispersion of an upscaled homogeneous model, however, assumes that the macro-dispersion process follows a Gaussian process (Dagan, 1990). It has been 455 shown here and in other work (e.g., Dagan, 1984; Cvetkovic et al., 1992) that solute transport in heterogeneous media instead produces significantly skewed plume distributions, with early peak of mass and a long tail. Approximating such a skewed distribution with a Gaussian curve that is located at the same center of mass travel time and has the same second spatial moment is know to generate earlier first travel times, a later peak of mass, and later late travel times, consistent with our results. This complexity of upscaling transport from heterogeneous conditions to a simplified homogeneous aquifer using lower spatial





moments only has been highlighted before. The results here confirm that for case of non-point source contaminations, but also
put the macro-dispersive process in relation to the well mixing process.

## 5.2    Time related capture zone

Analogous to the travel time $pdf$s, the spatial distribution of the stochastic capture zone, i.e., probability for a particle leaving a
given location of the NPS to reach a well, is highly impacted by the homogenization of the hydraulic conductivity, much more

so than by homogenization of land use processes alone (Figure 8).

In fully heterogeneous conditions, a wide range of $P_w$ values are distributed over a large portion of the domain surface.
However, most of the probabilistic capture zone is characterized by very low $P_w$ values. The critical zone (area of highest
probability) is characterized by $P_w$ values of $\approx$0.6 and is centered at a longitudinal distance of about $\approx$2000 meters from the
well. In the solution to the equivalent homogeneous parameter and boundary conditions, the uncertainty of the capture zone

location is significantly underestimated, with most of the capture zone being characterized by high probability to reach a well
(Figure 8b). Describing the spatial variability of nitrate mass loading and recharge (with an homogeneous $K$-field) only adds
a moderate degree of uncertainty to the capture zone delineation, as expected from travel time pdf results above. Utilizing the
alternative homogenized transport modeling approach with a homogenized $K$ and an equivalent macro-dispersion term, unlike
for travel time $pdf$s, does not substantially improve the stochastic prediction of the capture zone (compare Figure 8c and 8d).

Furthermore, homogenizing the hydraulic conductivity seems, independently of the description of the source terms or of the
consideration of dispersion, to mispredict the location of the capture zone: the critical zone is slightly moved downstream,
closer to the wells, and the capture zone extends to a small portion of the downstream edge of wells. The most distant part of
the critical zone in the homogenized prediction of the capture zone overlaps with the actual location of the critical zone in the
fully stochastic solution.

Consistent with these results, the spatial distribution of mean travel time required for the contaminant to reach a well is
similarly contracted to a much smaller area that extends downstream from the well, unlike in the fully stochastic representation
(Figure 9). The observed gradient of travel times, increasing with the distance from a well, is overestimated when $K$-fields are
upscaled. This leads to higher predicted mean travel times over the entire capture zone for all tested aquifer simplifications.

Simulation outcomes highlight that the set of upscaled $K$ values among the 50 realizations does not cover a range large

enough to reproduce the high variability of original contaminant location expected in heterogeneous situation. This indicates
that regional hydraulic vertical and longitudinal gradients, common to all simulations, control mostly the behavior of first
spatial moments of heterogeneous plumes used here to estimate apparent velocities. Thus, contaminant mass reaching the top
of the well has little variability – here only to the degree that the homogenization is done individually for each realization,
leading to some minor variability in the homogenized $K$ between realizations. More uncertainty is observed on the upstream

side of the capture zone since it represents mass reaching the bottom of the screen, the vertical position of which is realization
dependent.

Interestingly, the critical zone (high $P_w$) is predicted to be more downstream than its actual location if $K$ is homogenized
using apparent velocities. In case of heterogeneous $K$, a strong layering effect is observed, due to the superposition of relatively

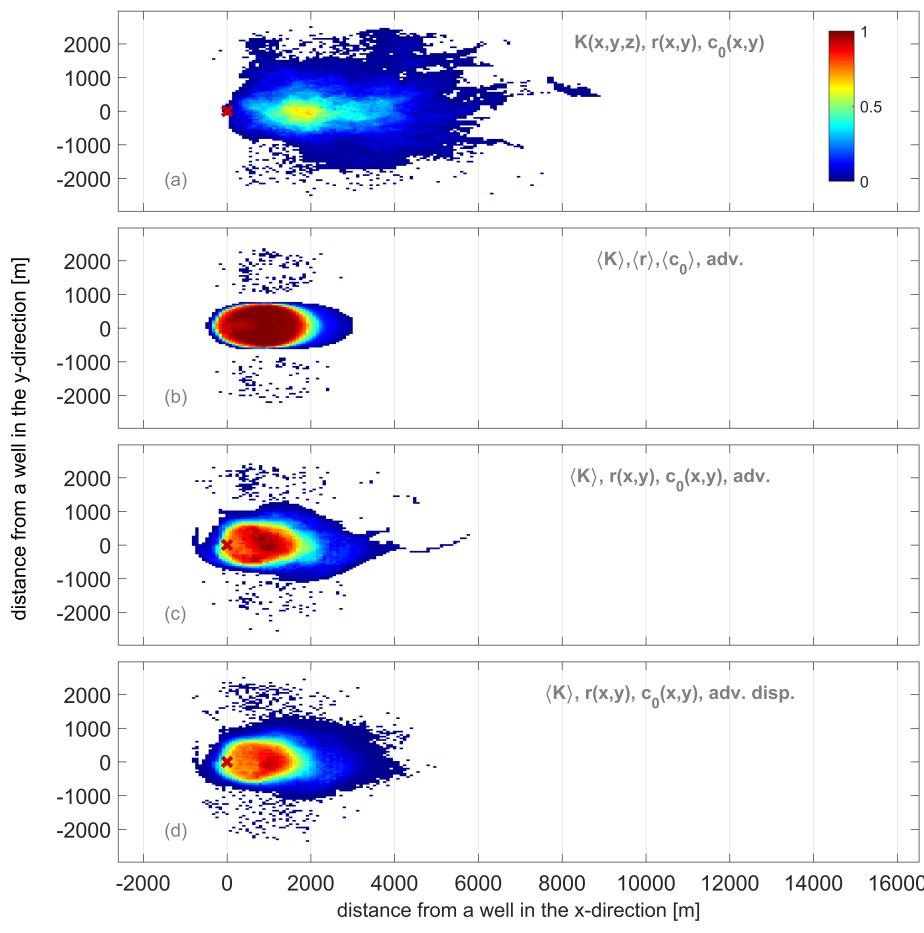

**Figure 8.** Probability of a particle leaving a given grid-cell to reach a well accounting for a heterogeneous $K$ field, heterogeneous $r$ and heterogeneous $c_0$ (reference case, a); an upscaled $K$ values, averaged $r$ and averaged $c_0$ accounting for advection only (b); an upscaled $K$ values, heterogeneous $r$ and heterogeneous $c_0$ accounting for advection only (c); an upscaled $K$ values, heterogeneous $r$ and heterogeneous $c_0$ accounting for advection and dispersion (d).



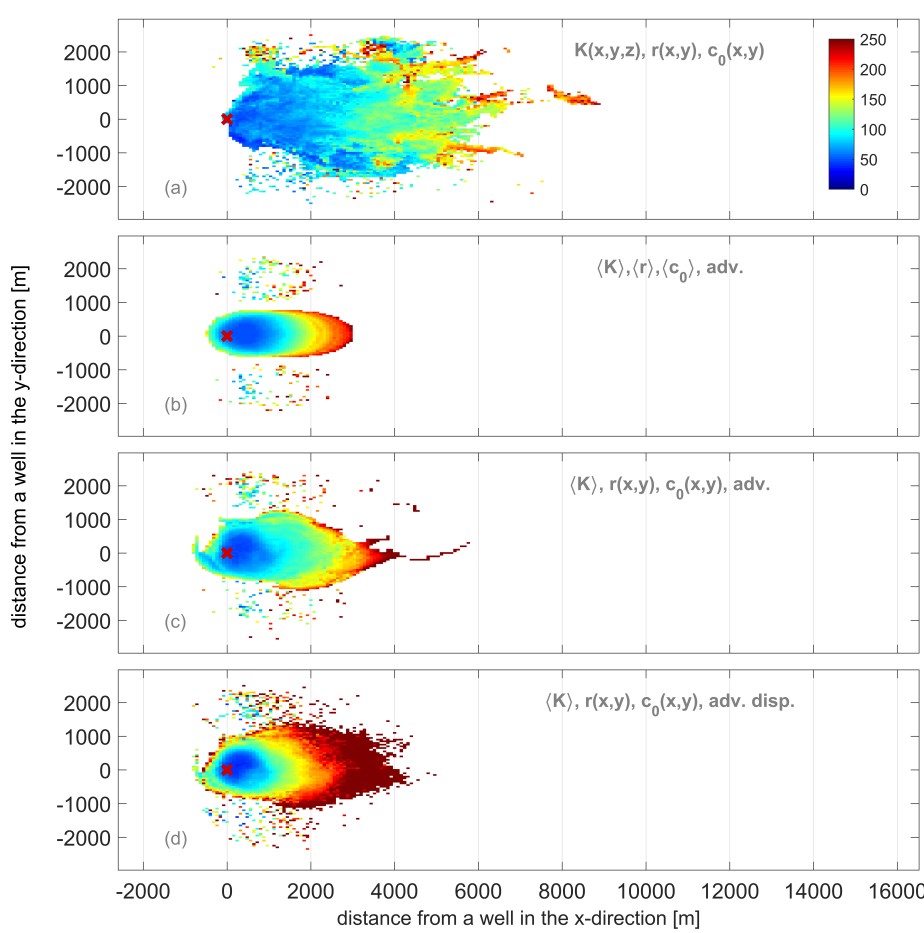

**Figure 9.** Expected travel time [years] of a particle leaving a given grid-cell and reaching a well for a heterogeneous $K$ field, heterogeneous $r$ and heterogeneous $c_0$ (reference case, a); an upscaled $K$ values, averaged $r$ and averaged $c_0$ accounting for advection only (b); an upscaled $K$ values, heterogeneous $r$ and heterogeneous $c_0$ accounting for advection only (c); an upscaled $K$ values, heterogeneous $r$ and heterogeneous $c_0$ accounting for advection and dispersion (d).





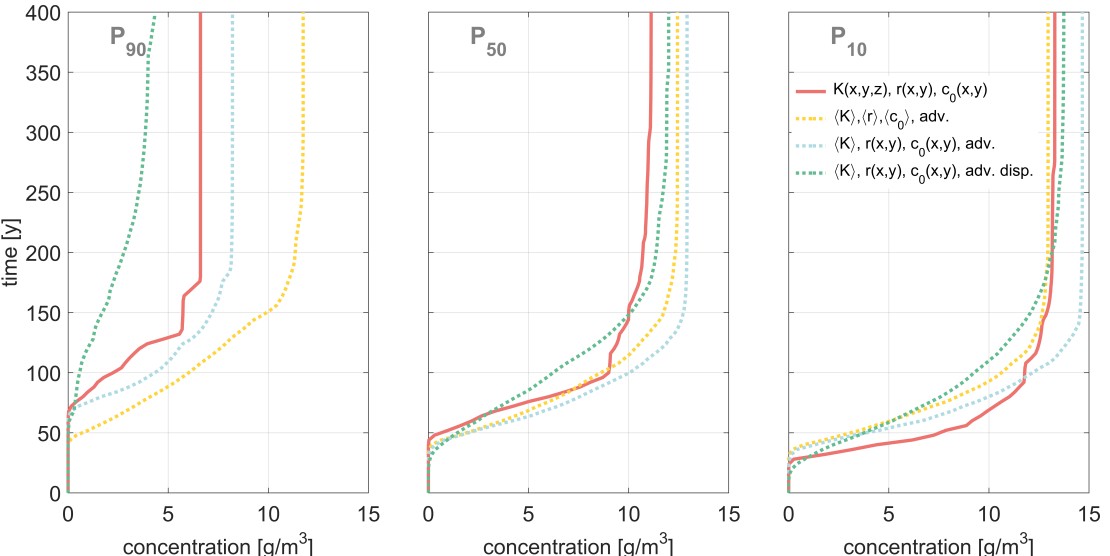

**Figure 10.** Time ($y$-axis) required for a well to exceed a given concentration ($x$-axis) with a probability of 90% (left), 50% (middle) and 10% (right) considering a heterogeneous $K$ field, heterogeneous $r$ and heterogeneous $c_0$ (reference case, red solid line); an upscaled $K$ values, averaged $r$ and averaged $c_0$ accounting for advection only (yellow dashed line); an upscaled $K$ values, heterogeneous $r$ and heterogeneous $c_0$ accounting for advection only (blue dashed line); an upscaled $K$ values, heterogeneous $r$ and heterogeneous $c_0$ accounting for advection and dispersion (green dashed line).

thin layers of highly and poorly conductive materials that stretch the plume longitudinally at large scale and move the capture
zone upstream.

### 5.3   Contaminant levels

Future concentrations exceeded by only 10% of wells ($P_{10}$) and those exceeded by half of wells ($P_{50}$) are captured to within a factor 2 for the transition period between 20 years and 150 years, but agree to within 10% with the fully stochastic simulation results at late time, under near steady-state pollution conditions. The (low) concentration levels exceeded by 90% of wells ($P_{90}$)
differs by a factor 2 or more, at all times, from the fully stochastic solution (Figure 10). Representing the spatial variability of source terms, but using a homogenized K-field improves the prediction of the $P_{90}$ evolution.

Using the alternative homogenized representation with an equivalent macro-dispersion improves the prediction only at late time ($> 150$ years) and predicts long-term concentrations for $P_{50}$ and $P_{10}$ very accurately (green lines in Figure 10). But it underestimates the concentrations for all of $P_{90}$ and during the transition time for $P_{50}$ and $P_{90}$.
The agreement between fully stochastic solutions and the homogenized solutions is in contrast to the seemingly significant differences between homogenized and fully stochastic results observed for travel time distributions of the individual mass percentiles and the capture zone location. That the homogenized prediction is still capable of producing useful results is due to





the unique properties of nonpoint source pollution listed earlier: First, the NPS pollution is a continuous process rather than a one-time event, with some interannual variability and slow long-term trends (Hansen et al., 2012; Harter et al., 2017). Second, the mixing of water quality occurring in the well screen greatly controls the observed pollutant levels because of the continuous loading and because differences in pollutant loading rates for the more permeable soils, across all crops (Table 5) vary within less then one order of magnitude. Third, the composition of the land use and therefore the recharge and mass loading rates vary at a scale that is much smaller than the source area of the well. Hence, any location of the source area will capture a similar overall mass of NPS pollutant over time. Third, the amount of water quality mixing in the well is strongly controlled by the vertical location and length of the well screen and, for typical municipal production wells or agricultural wells, as simulated here, well construction will dominate the range of travel time distributions water and solutes entering the well screen over effects of macro-dispersion. Reproducing the range of average regional gradients potentially observed in a region, and average loading therefore provides critical and important information to reproduce in-well mixing of age and, hence, recorded water quality.

Results show that homogenized $K$-fields perform more poorly to predict the lowest concentrations ($P_{90}$) than the highest ones ($P_{50}$ and $P_{10}$). From a NPS pollution management perspective, the accuracy of the higher concentrations exceeded by half of wells or even by just 10% of wells is most critical, since they are more likely to exceed the MCL. The homogenized predictions are least accurate during the transition (breakthrough) period when concentrations in the vertically mixed sample obtained from a well are strongly controlled by travel time $pdf$s, which in turn are affected by the heterogeneity in the land use and aquifer dynamics.

## 6   Results and Discussion: Regional stochastic analysis

Results and discussion thus far have focused on the uncertainty about predicting concentrations and source area associated with a single well, where land use distribution is heterogeneous, but deterministic (mappable). In the simulations discussed, the land use (but not the soil) was the same across all realizations. In NPS pollution management, an understanding of the variability in concentration evolution over time across the ensemble of wells in a basin, region, or management zone is of equal or more importance than understanding the uncertainty of future pollution dynamics at a particular well. For the regional analysis, the conceptual modeling approach is identical to the stochastic analysis of an individual well, except that the land use distribution is also a random variable. To adapt the simulation setup to the regional stochastic analysis, the spatial distribution of crops (i.e., land-use map) across the fixed grid of fields was randomly generated for each realization ("50 LUs"). Thus each realization represents an equi-probable location within a basin that is much larger in extent then the simulation domain. In the regional interpretation of the stochastic results, the range individual travel times, capture zones, and concentration breakthrough curves observed represent the variability across the ensemble of wells in the region, rather than the uncertainty about the outcome at a particular well (ergodicity principle, (Dagan, 1990)).

Adding random land use to the simulations leads to nearly identical travel time $pdf$s for early and median mass travel times appear and somewhat earlier late mass travel times (compare dashed and plain red lines in Figure 3). Travel times are



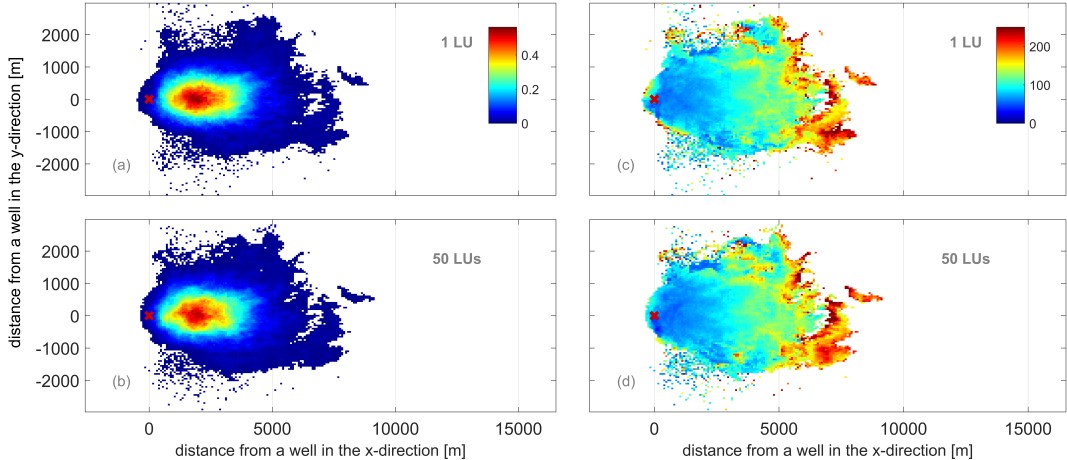

**Figure 11.** Probability of (left) and mean travel time required for (left) a particle leaving a given grid-cell to reach a well accounting for a single randomly generated land-use map for all realizations (top) and for a different randomly generated land-use map for each realization (bottom).

therefore largely insensitive to the stochastic conceptualization of the land-use spatial variability ("1 LU" compared to "50 LU"). Similarly, the capture zone area is not sensitive to whether a fixed heterogeneous ("1 LU") or random heterogeneous ("50 LU") stochastic concept is employed (Figure 11). As a result, lowest and highest contaminant levels ($P_{90}$ and $P_{10}$) are only slightly lower at late times, while $P_{50}$ are similar at any times for both analysis (compare dashed and plain red lines in Figure 6). The similarity in results here is due to the spatial scale of the land use variability, set by the size of the fields, with several dozen of fields occupying the critical area of the capture zone (see above). Given the mixing in the well screen and the continuity of NPS pollution, the number of fields in the capture zone is therefore sufficiently large, and the contrast in loading rates sufficiently small, that a single sample of the heterogeneous land use representation ("1 LU") becomes representative of an ensemble of land use patterns. That said, the advantages and disadvantages of the homogenization methods for land use and aquifer properties highlighted above apply equally to the depiction of regional variability in nitrate contamination of large production wells and to the uncertainty of nitrate dynamics in an individual well.

## 7   Conclusions

A significant body of groundwater flow and transport literature has focused on upscaling flow and transport processes associated with industrial point source pollution. For accidental pollution with pollutants exceeding compliance levels by orders of magnitude, field research has shown that large uncertainties exist in predicting the fate of such contaminant plumes and the inability of upscaled methods to capture site-specific plume behavior. Stochastic methods have been used to characterize such large uncertainties. Here we explore the ability to which homogenized, effective representations of aquifer structure and landscape spatial variability in flow and transport simulations of NPS pollution are capable of accurately predicting pollution





management metrics. We use three metrics typically of interest to NPS pollution management: travel time *pdf*s, stochastic capture zones, and stochastic breakthrough curves. We compare solutions of these metrics for a fully heterogeneous aquifer structure and landscape system with those of a homogenized, upscaled landscape system, those of a homogenized, upscaled aquifer system, and those of a completely homogenized aquifer and landscape system. Within the landscape system, we further distinguish between homogenizing recharge flux and homogenizing pollutant mass flux. The analysis is performed for a typical intensive, irrigated Mediterranean agricultural landscape of orchards, vineyards, and field crops overlying an alluvial aquifer system polluted with nitrate from fertilizer applications. Based on the simulation results presented, we make the following key conclusions:

- Land use, soil, and aquifer heterogeneity lead to large variability in groundwater travel paths, travel times, source location, and therefore well nitrate concentrations across a regional set of wells and, hence, significant uncertainty about pollution dynamics at any one well.

- The impact of continuous landscape pollutant loading to a typical high capacity production well with top of screen at 100 m below the water table is first seen a couple of decades after pollution initiation but is not fully reflected across all wells of a region after one to two centuries.

- With the capture zone of an individual well typically stretching across a diverse sub-set of land use in a region, the homogenization of the recharge and mass loading across the landscape to simulate NPS pollution management metrics can be appropriate, especially for simulating travel time *pdf*s and stochastic capture zones. In this case, nitrate variability between wells is much more affected by aquifer and soil heterogeneity than the heterogeneity in crop patterns across the landscape. This finding may not apply to cases where land use units (fields, orchards) occupy a much larger area or many fields of one crop type are clustered, or for wells with small pumping rates and, hence, small capture zones – in those cases the variability in capture zone loading across an ensemble of wells may be ill-represented by a homogenized, regionally averaged recharge and nitrate mass loading.

- Homogenization of the aquifer hydraulic property significantly degrades travel time statistics as well as the stochastic delineation of the capture zone. Accounting for aquifer heterogeneity by utilizing an upscaled macrodispersion only slightly improves predictions of travel time *pdf*s or stochastic capture zones.

- During the transition period (20 years to 170 years after pollution initialization), simulations using a homogenized representation of the aquifer structure provide aggregated concentration predictions, such as the concentration exceeded by half of wells, that are as much as a factor 2 different from predictions that fully represent aquifer heterogeneity.

- On the other hand, due to the strong effect of vertical groundwater mixing during the well pumping process and due to the continuity of NPS pollution, an upscaled, homogenized representation of aquifer heterogeneity using an effective hydraulic conductivity produces reliable and useful predictions for the concentration levels exceeded by half of wells and even the higher concentrations exceeded by only 10% of wells, especially in the long-term. These are the wells of most concern in NPS pollution of groundwater.





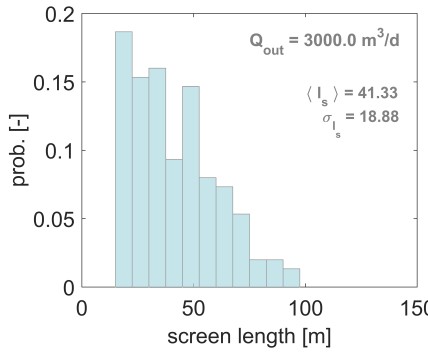

**Figure A1.** Probability histogram of the simulated screen lengths.

- – Homogenized approaches may be most useful to predict whether long-term outcomes meet management goals across a regional ensemble of wells, but may be less accurate in specifying how quickly such goals may be achievable.

Future work is needed to further understand the role of crop type clustering on landscape homogenization, and the effect of interannual and seasonal loading variability on NPS pollution management metrics. More work is also needed to investigate other forms of partial or full homogenization of aquifer structure on prediction metrics considered here.

*Code availability.* The online resources located in GitHub (see this link) include the Matlab scripts necessary to reproduce the results.

## Appendix A: Well screen design

For each realization of the hydraulic conductivity field, 3 extraction wells are implemented. The pumping rate of each well is fixed to 3000 m$^3$/d and the top of the screen is fixed to 100 m. As in real settings, the length of this screen is dependent on the local aquifer properties in order to sustain the total extraction rate. Indeed, pumping effectively occurs through portions of well located in highly conductive aquifer material. To simulate this local K dependence of the well screen length, we are using a rule of thumb stating that 10 cumulative foot (3.05 m) of gravel and sand has to be crossed for each 100 gallon-per-minute (545.1 m$^3$/d) of extraction. The probability histogram (over all realizations) of the simulated screen lengths for each tested extraction rates is shown in Figure A.1.

## Appendix B: Impact of dispersion

Former studies (LaBolle, 1999; LaBolle and Fogg, 2001; Weissmann et al., 2002) highlighted the insensitivity of transport simulations to local scale dispersivity ($\alpha_i$, where $i$ indicates the transport direction) if aquifer heterogeneity is representing in a finely detailed manner by means of the transition probability method (TPROGS). This insensitivity is explained by the large





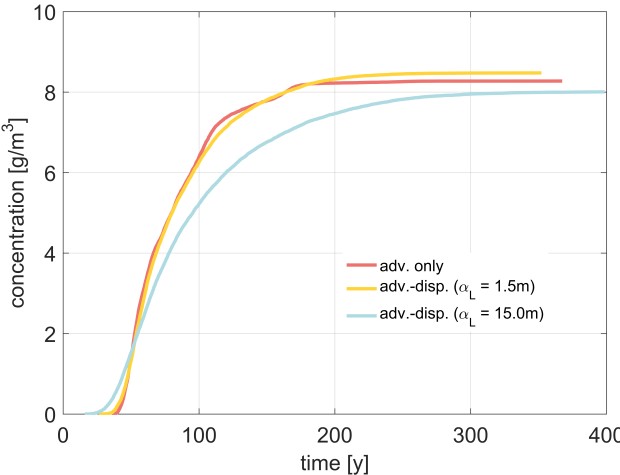

**Figure B1.** Breakthrough curve recorded at a well accounting for advection only (red cruve), for advection and dispersion with a longitudinal dispersivity of 1.5 m (yellow curve), and for advection and dispersion with a longitudinal dispersivity of 15.0 m (blue curve). Transverse horizontal and transverse vertical dispersivities are always, respectively, 1/10 and 1/100 of the longitudinal dispersivity.

macrodispersion caused by the well represented facies scale heterogeneity, which renders spreading from local dispersivity insignificant. As a result, fairly small values of $\alpha_L$ are, in this setting, usually adopted. For instance, Weissmann et al. (2002) applied to their transport model (with computational grid similar to the one used in our study) a grid scale longitudinal dispersivity of 0.04 m, which appeared to have a insignificant impact on transport and resulting groundwater age distribution. Values of $\alpha_L$ were chosen to fulfill the magnitude of dispersivity values reported at field sites of scale similar to the computation cells

(references in previously cited work). Here, we test the impact of much larger values of dispersivity (1.5 m and 15.0 m) on breakthrough curves recorded at a well. The simulation setup is identical to the one described in the manuscript. Results are shown for a single realization of the hydraulic conductivity field.

Our outputs displays no significant impact on transport of a $\alpha_L$ coefficient of 1.5 m . Increasing grid-scale dispersivity to 15.0 m leads to slightly earlier first travel times, later late travel times and lower contaminant levels observed at intermediate

and late times (Figure B1). Therefore, no implications can be expected when accounting exclusively for advection when grid-dispersivity is lower than 1.5 m, as always used in previously cited studies.

*Author contributions.* CVH and TH designed the Monte Carlo simulations, which was then ran and post-processed by CVH. ED designed and ran the Hydrus simulations. All authors contributed to the results analysis and to the writing of the article.

*Competing interests.* The authors declare that they have no conflict of interest.



*Acknowledgements.*   The authors gratefully acknowledge the financial support through the California State Water Resources Control Board (SWB), Agreement 15-062-250. Any use of trade, firm, or product names is for descriptive purposes only and does not imply endorsement by the SWB.





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
