# Peer review of "On the Conceptual Complexity of Non-Point Source Management: Impact of Spatial Variability"

_Hydrology and Earth System Sciences, 2019_

## Referee Comment (RC1) · Anonymous Referee #1 · 1 Nov 2019

Overview:

This is an interesting work that brings together many contributions in the field of probabilistic risk assessment (PRA) in aquifers to investigate transport of non-point sources (NPS). The authors explore parameters such as recharge rates and contaminant loadings in the final model output. Furthermore, the authors attempt to reduce the complexity of the model by upscaling a set of spatially/temporally variable quantities, such as the hydraulic conductivity, on the management of NPS. Through the use of numerical simulations, the authors provide an analysis that couples vadose zone, aquifers and land use in a single framework.

[Figure]

I enjoyed reading this paper given that it aims in bringing in elements of stochastic hydrogeology into decision making. The material is well written and organized. The illustrations are clear and well depicted. The referencing is also appropriate although some contributions in the PRA of contaminated aquifers are missing. This is not a big issue. Through the use of scaling arguments (i.e. compliance planes, source sizes etc) the authors make a compelling argument to evoke upscaling for the problem at hand. They claim that due to significant mixing in the compliance plane and the lack of significant variability in NPS solutes, the uncertainty in predictions are reduced thus leading more simplified approaches for modeling such complex systems. Results indicate that the mass arrival time distributions are not that sensitive to the spatial variability of recharge and solute loading whereas some sensitivity is observed for the concentration signal and capture zone estimation. The authors also show that homogenization of the conductivity affects the uncertainty of arrival times.

Specific comments:

-The authors refer to the word ergodicity multiple times. Ergodicity in what? I think they are referring to ergodicity in the transport behavior. If so, provide a quantitative measure of what ergodicity is. For example, the ratio between the source zone dimension and correlation scale needs to be large. If this is the case, than why one would need to quantify uncertainty due to the conductivity field? The spatial statistics is representative of the ensemble statistics. This needs to be better discussed.

-I am not sure if I missed this in the text but it would be interesting to see if the upscaled dispersion reaches its Fickian limit. Looking at figure SM6, it seems that this is not the case and therefore, transport is still subject to uncertainty. To my understanding, based on the histograms, these upscaled dispersion coefficients reported in figure SM6 are not the ones in the Fickian limit and therefore ergodicity is not attained. So how is it that the authors claim "ergodicity" in this paper?

-It would be interesting to see how the conclusions regarding recharge reported in this

paper compare with the ones reported in the works of Rubin and Bellin (1994) WRR and Li and Graham (1999) WRR. These authors investigate the impact of recharge and its randomness on travel time pdfs.

-Line 460: "The results here confirm that..., but also put the macro-dispersive process...". I could not understand the meaning of this sentence. Please revise its structure. Thanks.

---

## Referee Comment (RC2) · Anonymous Referee #2 · 8 Nov 2019

The present work deals with the feasibility of simplify the conceptual modelling of non-point source (NPS) contamination of an aquifer, in the presence of an active pumping well. The simplification consists of spatial homogenization of otherwise heterogeneous terms (recharge rate and contaminant concentration of the recharging water) and of the aquifer hydraulic properties (conductivity). I liked the general goal/purpose of the study (even though there is no theoretical/technical novelty in the employed methodology) and its practically oriented nature. Nevertheless, it is my opinion that there are several unclear points and unstained observations that should be addressed before acceptance for HESS. I list my concerns below. I do see a very good paper after addressing these issues/confusions. I also hope that my comments help in making the

paper shorter and clearer, it becomes quite hard to follow it from the beginning to the end.

Line 5-9: 'On the other hand, concentration levels of some key NPS contaminants (salinity, nitrate) vary within a limited range (<2 orders of magnitude); and significant mixing occurs across the aquifer profile along the most critical compliance surface: drinking water wells with their extended vertical screen length. Here, we investigate, whether these two unique NPS contamination conditions reduce uncertainty such that simplified spatiotemporal representation of recharge and contaminant leakage rates and of hydraulic conductivity are justified when modeling NPS pollution.' I don't agree with the fact that the Authors tested that the two mentioned conditions imply a reduction in the uncertainty (they do not explore scenarios where these two conditions aren't meet!). I would say that the Authors investigate, under these two peculiar NPS conditions, the possibility of introducing simplifying modelling hypotheses which do or do not affect the ensuing uncertainty about targeted quantity. Please consider revise the sentence, here and throughout the whole paper where needed.

Line 14-15: 'Surprisingly, regional statistics of well concentration time series are fairly well reproduced by a series of equivalent homogeneous aquifers, highlighting the role of NPS solute mixing along well screens.' I have two comments here: (i) the term regional statics is somewhat obscure; (ii) once clarify it in the section 6 (see also my comment 3), it is my understanding that the diverse homogenization here proposed work fine for low (i.e., 0.1) and middle (i.e.,0.5) probability of exceeding a given concentration in the well, while for high probability (i.e., 0.9) the homogenization-based results are unreliable (see e.g., the discrepancy between the curves in Figure 6a and Figure 10a, in Fig. 10a I don't see the 'regional' counterpart (red dashed curve)).

Lines 90-92: 'Assuming ergodicity (Dagan, 1990), stochastic management metrics are quantified both, for the pollution variability across an ensemble of production wells encountered over a basin, and for the uncertainty about pollution levels at an individual well.' As a matter of fact the Author verified the validity of the ergodicity principle

(here in the sense that the results of one single realization are representative of the results across the whole ensemble of realizations, or, in other words, there is no variability across the ensemble of realizations of the investigated output) only with respect to the soil crop arrangement (see Fig. 1), but ergodicity does not hold with respect to the (either heterogeneous or homogenized) aquifer conductivity (i.e., their results clearly show that there is a variability in the investigated NPS management metrics as the conductivity distribution varies among realizations)! Please clarify this point. The Authors named the regional analysis ('pollution variability across an ensemble of production wells encountered over a basin') (see also Section 6) the scenario in which both the conductivity and the land crop usage varies between the Monte Carlo simulations. They referred to the single well analysis ('uncertainty about pollution levels at an individual well') when only the conductivity varies among the MC simulations (with the crop arrangement fixed). To me this distinction is not meaningful, since it implies that in the groundwater basin there are sub-portion (the simulated domain) subjected to the very same boundary conditions (aside from the infiltration rate) and that these sub-portions do not influence each other (they are far away from one another). In my vision the Authors have, given a domain of interest and deterministic initial and boundary conditions (aside from the infiltration rate), conducted the uncertainty analysis (i) for a given (i.e., conditional) to a crop arrangement and (ii) considering the uncertainty in the crop arrangement, proven then that the uncertainty in the latter is not an influential factor (i.e., there is ergodicity w.r.t. to the crop usage). Please consider this aspect, the regional and one-well distinction seems to me confusing and not well supported by the investigated set-up.

Line 115-117: 'Assuming ergodicity (Dagan, 1990), stochastic analysis is applied to first quantify uncertainty about pollution outcomes at individual wells and to secondly quantify regional spatial variability in pollution outcomes across an ensemble of wells'. Please revise this sentence according with Comment 2-3. Furthermore, in the way it is written it means that ergodicity is needed in order to quantify uncertainty (this is the case for example in geostatistical approach where spatially distributed measure of conductivity in a field, i.e., in a single realization, are employed to describe the ensemble statistic of conductivity) in this study, i.e., that automatically a single realization is sufficient to describe the behavior of the ensemble, whereas, it is my understanding that the Author did the other way around: prove the validity of ergodicity (for the investigated quantity) w.r.t. to the crop usage. If ergodicity was originally assumed, no need to do many simulations.

Line 129-130: 'The histograms of the mean and the variance of the logarithm of K are shown in Figure SM3. Fifty realizations were sufficient to converge the lower statistical moments of K and of the resulting mean velocities (Figure SM7)'. It is not clear at all that the Authors are referring to the spatial mean and spatial variance (this is my impression) evaluated for each field of K, and then doing the histogram of these quantity. Is it so? Why do we care about it? How do the Author prove the convergence of these spatial mean and spatial variance employing 50 realizations? To which values should these quantities converge? I am way more concerned about: are the 50 realizations enough to ensure the convergence of the statistics (e.g., pdf, CDF) of the output quantities of interest (i.e., travel time, breakthrough curves at the well and capture zone)? This is aspect is not investigated at all by the Authors and looking to the high spatial variance of K (from 10 to 18 approximately) I am afraid that 50 realizations are not enough, even if the source is spatially distributed. Please analyze the convergence of the investigated results w.r.t. to the Monte Carlo simulations. Furthermore, regarding the histogram in SM3 and SM4 for the recharge rate and the concentration of pollutant in the recharge, why do we care about them? Please clarify.

Line 151-195: I see the detailed description of the estimation of the recharge rate to better fit in an Appendix.

Line 230-233: 'The detailed discretization of the velocity field described above is capturing the most relevant characteristics affecting the macro-dispersive transport behavior (LaBolle, 1999; LaBolle and Fogg, 2001; Weissmann et al., 2002; Henri and Harter, 2019). Therefore, effects of grid-scale dispersion are assumed to be negligible.'

So, does the Author set the tensor D in (10) equal to zero? Please clarify.

Line 244-246: 'Three relevant nonpoint source (NPS) pollution management metrics are considered to measure the stochastic simulation outcomes: the probability distribution of pollutant travel times to wells, the probability distribution of pollutant concentration in wells, and the probability distribution of source locations.' These are the 3 quantities of interest, why do you introduce them in the 2.3.1 Pollutant travel times section? Better just before.

Line 250-251: 'Following a stochastic approach, probability density functions (pdfs) of travel times $t_i$ are obtained by determining the histogram of $t_i$ in 150 simulated wells'. Please note that the histogram is not a pdf, whereas the latter is associated with a continuous variable and the former to a discrete variable. I would limit to say to that the pdf are estimated on top of the 150 simulated wells. Furthermore, at lines 244-246 the probability distribution is mentioned, this is not the pdf. Also, at line 214 the authors say that they analyzed the probability distribution, but then in Figure 3 they depict the pdf. Please check for the consistency of the terminology/results through all the work.

Lines 262-263: 'NPS pollution management may also require the assessment of the effective source area, i.e., the capture zone or contributing area of the pollution observed in a production well.' Please avoid to use source zone, this is typically used to indicate the area covered by the contaminant at the initial time (regardless if it reach the well or not), the capture zone of the well is way more clear as the wording in my opinion.

Line 275-277: 'The NPS metrics from fully heterogeneous simulations are compared to the NPS metrics obtained from a range of upscaled, homogenized simulations that employ effective homogeneous properties rather than the original heterogeneous distribution of the K, r, and $c_0$ terms'. Please consider avoiding the word upscale/upscaling (here and in the whole text), since this inherently implies a change of scale (e.g., from pore to continuum or from continuum to continuum) and it is typically associated with a

change of the governing equations used to describe the process (e.g., effective model for the solute transport involving non-local terms) whereas here the Author conducted a simple homogenization (with the arithmetic average as a rule) of the diverse terms.

Line 286-292: 'To simulate flow and transport in an equivalent homogeneous, upscaled K conditions, we estimate the effective longitudinal and transverse vertical hydraulic conductivity, $K^*x$ and $K^*z$, and dispersion, $alpha^*L$ and $alpha^*T$. Effective parameters in the longitudinal direction ($K^*x$ and $alpha^*L$) are determined from the first and second spatial moments of a plume resulting from an injection of mass in a vertical plane of width 3000.0 m and depth 50.0 m. The same approach is adopted to estimate the effective parameters in the transverse vertical direction ($K^*z$ and $alpha^*T$) by injecting particles in a horizontal plane covering the entire top of the domain. No extraction is considered in both cases in order to capture the natural behavior of the plume.' What about the $K^*y$ and dispersivity in the y direction? Furthermore, 'No extraction is considered in both cases in order to capture the natural behavior of the plume', but the ensuing macro-dispersion of a plume is influenced by the presence of pumping well (e.g., radial versus uniform flow conditions), please clarify/justify this aspect.

Lines 317-316: 'For all simulations, early mass travel times are within a range of 10 to 100 years with an expected value (highest probability) of 50 years (Figure 3a).' For a continuous variable is not possible to define the value with the highest probability (it is possible for a discrete variable). Furthermore, the expected value of a pdf does not always coincide with the value where the value of the pdf is the highest. Clarify.

Lines 346-347: 'Spatial variability in the recharge is responsible for somewhat more uncertainty in the exact delineation of the capture zone along its margins than what is captured by the homogenization of r.' Is not r the symbol to indicate the recharge? I think r should be replaced with C0 or vice-versa. Furthermore, focusing on the impact of the homogenization of the recharge (compare Fig. 4a with Fig. 4c) I would say that the homogenization of the latter is associated with a less spatially extended capture zone w.r.t. to the case in which the recharge is treated as heterogeneous, this does

not mean that the there is more uncertainty in the delineation of the capture zone. Furthermore, looking to the Fig. 4 I would say that capture zone is not well-delineated along its border, due to the low number of Monte Carlo realizations. Please check it and revise the sentence.

Lines 379-380: 'For an individual well, the results indicate that there is a 10% chance for nitrate concentrations to start to rise before 30 years, a 50% chance to rise no later than 50 years, and a 90% chance to rise before 70 years'. Should not be after X years?

Lines 402-404: 'Homogenizing only concentration also leads to an underprediction, by about 20%, of concentrations exceeded by either 90%, 50%, or 10% of well, relative to the fully stochastic land use treatment (green lines in Figure 6).' Is green correct?

Lines 471-472: ' ... only adds a moderate degree of uncertainty to the capture zone delineation..' see Comment 15.

Line 480: 'The observed gradient of travel times' I don't see any gradient (i.e., âĹĞ) of the travel times evaluated by the Authors. There is a spatial variation of the mean travel time in Fig. 9, but this is different from a proper evaluation of the gradient. Please revise the sentence

Lines 487-489: 'Thus, contaminant mass reaching the top of the well has little variability – here only to the degree that the homogenization is done individually for each realization, leading to some minor variability in the homogenized K between realizations.' Why does the fact that the mass reaching the top of the well exhibits a low variability (note that the Authors do not provide a quantification of it) lead to have minor variability in the homogenized K? Please clarify.

Lines 489-491: 'More uncertainty is observed on the upstream side of the capture zone since it represents mass reaching the bottom of the screen, the vertical position of which is realization dependent' I suppose this comment is related to Fig. 9, which depicts the expected value of the travel time to the well. The latter gradually increases

as we move upstream w.r.t. to the well location. I don't see this as a measure of an increasing level of uncertainty! It could be that the expected value of the travel time increases, but it can also be that spread (e.g., measured through the variance) of the pdf decreases. Please consider evaluating at least the variance of the pdf of travel times as a quantification of the degree of uncertainty.

Lines 520-521: 'Results show that homogenized K-fields perform more poorly to predict the lowest concentrations (P90) than the highest ones (P50 and P10).' Why P90 is associated with low concentrations? P90 is a probability, looking at Fig. 10a I see that for a given concentration (either high or low) there is a time necessary to exceed this level of concentration in the well with a probability of 0.9. The way I interpret Fig. 10 is that homogenized solutions are in good agreement with the heterogeneous case, when it is a matter of evaluating low (P10) and middle (P50) probability of exceedance of a given concentration, but the homogenized solutions do not work well for high (P90) probability of exceedance of whatever concentration.

Conclusion: I note a change of style in the conclusion, whereas there is a more consistent and proper use of the terminology with respect to the rest of the work.

––––––––––––––––––––––––

---

## Author Comment (AC2) · 20 Dec 2019

**Referee #2 Evaluations**

The present work deals with the feasibility of simplify the conceptual modelling of non- point source (NPS) contamination of an aquifer, in the presence of an active pumping well. The simplification consists of spatial homogenization of otherwise heterogeneous terms (recharge rate and contaminant concentration of the recharging water) and of the aquifer hydraulic properties (conductivity). I liked the general goal/purpose of the study (even though there is no theoretical/technical novelty in the employed method- ology) and its practically oriented nature. Nevertheless, it is my opinion that there are several unclear points and unstained observations that should be addressed before acceptance for HESS. I list my concerns below. I do see a very good paper after addressing these issues/confusions. I also hope that my comments help in making the paper shorter and clearer, it becomes quite hard to follow it from the beginning to the end.

> Thank you for your constructive and well-thought-out comments. We appreciate the attention to the details of our work. Please see below our answers (in blue) to your concerns. Proposed additions to, deletions from, or modification of the manuscript are written in green.

- Line 5-9: *'On the other hand, concentration levels of some key NPS contaminants (salinity, nitrate) vary within a limited range (<2 orders of magnitude); and significant mixing occurs across the aquifer profile along the most critical compliance surface: drinking water wells with their extended vertical screen length. Here, we investigate, whether these two unique NPS contamination conditions reduce uncertainty such that simplified spatiotemporal representation of recharge and contaminant leakage rates and of hydraulic conductivity are justified when modeling NPS pollution.'*
  I don't agree with the fact that the Authors tested that the two mentioned conditions imply a reduction in the uncertainty (they do not explore scenarios where these two conditions aren't meet!). I would say that the Authors investigate, under these two peculiar NPS conditions, the possibility of introducing simplifying modelling hypotheses which do or do not affect the ensuing uncertainty about targeted quantity. Please consider revise the sentence, here and throughout the whole paper where needed.

  > Thank you for making this excellent point. To clarify our approach, we propose to change this sentence to:

  > *"Given these two unique NPS contamination conditions, we here investigate the degree to which NPS travel time to wells and the NPS source area associated with an individual well can be appropriately captured, for practical applications, when spatiotemporally variable recharge, contaminant leakage rates, or hydraulic conductivity are represented through a sub-regionally homogenized parametrization." (line 7)*

- Line 14-15: '*Surprisingly, regional statistics of well concentration time series are fairly well reproduced by a series of equivalent homogeneous aquifers, highlighting the role of NPS*

*solute mixing along well screens.'*

I have two comments here: (i) the term regional statistics is somewhat obscure; (ii) once clarify it in the section 6 (see also my comment 3), it is my understanding that the diverse homogenization here proposed work fine for low (i.e., 0.1) and middle (i.e.,0.5) probability of exceeding a given concentration in the well, while for high probability (i.e., 0.9) the homogenization-based results are unreliable (see e.g., the discrepancy between the curves in Figure 6a and Figure 10a, in Fig. 10a I don't see the 'regional' counterpart (red dashed curve)).

The term regional is not necessary here. We propose to delete that in a revised version. And indeed, the concentrations with a high probability to be exceeded are not well reproduced if K is homogenized. Yet, these lower concentrations are rarely relevant in nonpoint source contamination management. The last sentence of the abstract will be rewritten to clarify this point:

*"Surprisingly, the statistics of relevant NPS well concentrations (fast and intermediate travel times) are fairly well reproduced by a series of equivalent homogeneous aquifers, highlighting the role of NPS solute mixing along well screens." (line 14)*

- Lines 90-92: *'Assuming ergodicity (Dagan, 1990), stochastic management metrics are quantified both, for the pollution variability across an ensemble of production wells encountered over a basin, and for the uncertainty about pollution levels at an individual well.'* As a matter of fact the Author verified the validity of the ergodicity principle (here in the sense that the results of one single realization are representative of the results across the whole ensemble of realizations, or, in other words, there is no variability across the ensemble of realizations of the investigated output) only with respect to the soil crop arrangement (see Fig. 1), but ergodicity does not hold with respect to the (either heterogeneous or homogenized) aquifer conductivity (i.e., their results clearly show that there is a variability in the investigated NPS management metrics as the conductivity distribution varies among realizations)! Please clarify this point.

We agree that indeed the paper shows ergodicity w.r.t. to crop coverage, that is, ergodicity in the sense that the crop variability across the source area is sufficiently large that any realization of crop produces the same outcome (for the stochastic management metric).

But we consider ergodicity here also in a different context. The issue has been raised by reviewer #1. So, we here copy our response to reviewer #1 as this should address reviewer #2's concern as well:

Thank you for raising your concern about ergodicity in these two comments. The context in which the term was used in the manuscript was, indeed, unclear.

The assumption of ergodicity is here employed NOT in the sense that we assume the NPS plume to be so large that there is no variability in plume moments between realizations. Instead, we here employ the ergodicity hypothesis in the same way as described, e.g., by Rajaram (2002) or Gelhar (1993). There, a single large realization (spatially extending over many correlation scales) of the K field is used to compute sample moments (e.g., of head, concentration). The ergodic hypothesis is employed to justify comparing these sample moments to the theoretical ensemble moments obtained analytically. But the ensemble moments also characterize the uncertainty about, e.g., head or concentration at a single unmeasured location within the aquifer represented by the simulation.

Analogous, we here use the ergodic hypothesis to postulate that the statistical sampling across the 150 well and 150 source areas simulated is representative of the ensemble

moments (the stochastic management metrics) at an individual unmeasured well and its source area in the real-world aquifer represented. The 150 simulated wells and source areas are analogous to the single large realization in Rajaram (2002), since the K field is stationary.

In other words, we must assume ergodicity and stationarity to be able to equate the stochastic NPS management metrics obtained from across 150 samples (wells, source areas) to their ensemble properties. The ensemble properties in turn apply to any (unmeasured) well and its source area in the aquifer system. They characterize the uncertainty about these metrics at each (unmeasured) location (well, source area).

We also refer to the histogram of the mean and variance of the K fields in supplementary material (Figure SM3, to be updated in the new version of the manuscript and shown below) showing stationarity in the moments of the K field, i.e., a very narrow range of values for the mean and variance, and therefore displaying that structural ergodicity can be assumed.

[Figure]

Figure SM3: Histogram of the mean (top) and variance (bottom) of the log-normal hydraulic conductivity.

Reference: *Rajaram (2002), Perturbation theories for the estimation of macrodispersivities in heterogeneous aquifers, in Rao S. Govindaraju, Stochastic methods in subsurface contamination hydrology, p13-62.*

*Gelhar, L. W. (1993), Stochastic subsurface hydrology: Englewood Cliffs, New Jersey, Prentice-Hall, 390 p.*

The Authors named the regional analysis ('pollution variability across an ensemble of production wells encountered over a basin') (see also Section 6) the scenario in which both the conductivity and the land crop usage varies between the Monte Carlo simulations. They referred to the single well analysis ('uncertainty about pollution levels at an individual well') when only the conductivity varies among the MC simulations (with the crop arrangement fixed). To me this distinction is not meaningful, since it implies that in the groundwater basin there are sub-portion (the simulated domain) subjected to the very same boundary conditions (aside from the infiltration rate) and that these sub-portions do not influence each

other (they are far away from one another). In my vision the Authors have, given a domain of interest and deterministic initial and boundary conditions (aside from the infiltration rate), conducted the uncertainty analysis (i) for a given (i.e., conditional) to a crop arrangement and (ii) considering the uncertainty in the crop arrangement, proven then that the uncertainty in the latter is not an influential factor (i.e., there is ergodicity w.r.t. to the crop usage). Please consider this aspect, the regional and one-well distinction seems to me confusing and not well supported by the investigated set-up.

Indeed, we assume that there are sub-portions of the aquifer that do not influence each other. The size of the simulation domain was chosen to fully accommodate the source area of three production wells. Since we show that their source areas do not overlap, each well's area of capture can indeed be considered independent.

The reviewer is also correct that we make a conceptual distinction between the case of assessing the uncertainty about an individual well's source area and contaminant travel time, and the case of assessing the variability of source area and contaminant travel time of many wells in a region with a stationary hydraulic conductivity, recharge, NPS mass loading, and landuse pattern. We argue that this is an important theoretical distinction, for the same reason that the concept of ergodicity is relevant.

From a theoretical standpoint, therefore, generating 50 realizations (each with 3 wells), without changing actual landuse is, in principal, only assessing the uncertainty about recharge, loading, and K under a given spatial configuration of landuse (which would be specific to each of the three wells).

Then also making landuse spatial pattern a stochastic variable is conceptually consistent with the idea of simulating 150 different wells, each with its own spatial landuse arrangement.

But the reviewer is correct in pointing out that there is negligible difference between these two, effectively proving ergodicity w.r.t. landuse. But we do not believe it is useful to say this explicitly (e.g., in a sentence at the end of section 6), because we don't want to confuse the reader with the "other" (valid) interpretation of ergodicity that the reviewer is in fact employing here.

We therefore propose to add the word "pattern" to the line 533, make a small change in line 536

*"To adapt the simulation setup to the regional stochastic analysis, the spatial distribution pattern of crops…." (line 533)*

*"…… interpretation of the stochastic results, the range OF individual travel times, capture zones, …." (line 536, capitalized word to be inserted)*

- Line 115-117: *'Assuming ergodicity (Dagan, 1990), stochastic analysis is applied to first quantify uncertainty about pollution outcomes at individual wells and to secondly quantify regional spatial variability in pollution outcomes across an ensemble of wells.'*
  Please revise this sentence according with Comment 2-3. Furthermore, in the way it is written it means that ergodicity is needed in order to quantify uncertainty (this is the case for example in geostatistical approach where spatially distributed measure of conductivity in a field, i.e., in a single realization, are employed to describe the ensemble statistic of conductivity) in this study, i.e., that automatically a single realization is sufficient to describe the behavior of the ensemble, whereas, it is my understanding that the Author did the other

way around: prove the validity of ergodicity (for the investigated quantity) w.r.t. to the crop usage. If ergodicity was originally assumed, no need to do many simulations.

See our answer to the previous two comments.

- Line 129-130: *'The histograms of the mean and the variance of the logarithm of K are shown in Figure SM3. Fifty realizations were sufficient to converge the lower statistical moments of K and of the resulting mean velocities (Figure SM7)'.*
It is not clear at all that the Authors are referring to the spatial mean and spatial variance (this is my impression) evaluated for each field of K, and then doing the histogram of these quantity. Is it so? Why do we care about it? How do the Author prove the convergence of these spatial mean and spatial variance employing 50 realizations? To which values should these quantities converge? I am way more concerned about: are the 50 realizations enough to ensure the convergence of the statistics (e.g., pdf, CDF) of the output quantities of interest (i.e., travel time, breakthrough curves at the well and capture zone)? This is aspect is not investigated at all by the Authors and looking to the high spatial variance of K (from 10 to 18 approximately) I am afraid that 50 realizations are not enough, even if the source is spatially distributed. Please analyze the convergence of the investigated results w.r.t. to the Monte Carlo simulations. Furthermore, regarding the histogram in SM3 and SM4 for the recharge rate and the concentration of pollutant in the recharge, why do we care about them? Please clarify.

As mentioned before, the histogram of the mean and variance of the lnK fields were erroneous (they were considering the first layer of the domain only). Please apologize the mistake. Below are the corrected histograms, which provide a better measure of the high degree to which the entire range of K variability is captured in each of the 50 realizations.

The convergence of the apparent velocities and of apparent dispersivities, which are derived from particle plumes, give an indication about convergence of transport metrics. The dataset used to compute statistics is composed of 150 values of well concentrations and of travel times (3 wells per realization). For the sake of clarity, we now also show in Figure SM6 and SM7 (see below) the convergence of the mean and variance of the travel times and of the 50% highest concentrations, respectively. We propose to add the following text to the manuscript:

*"Figure SM6 shows satisfactorily the convergence of the mean and variance of $t_{50}$."* (line 252)

*"Figure SM7 shows satisfactorily the convergence of the mean and variance of the concentration exceeded by 50% of wells."* (line 259)

The histograms for the recharge rate (SM4) and the input concentrations (SM5) are given in supplementary material in case a reader may have a concern about the range of values used. The following text would be added to the manuscript:

*"For indications about the range of values and degree of variability, histograms of the mean and variance of the recharge rates applied in the 50 realizations of heterogeneous cases are shown in Figure SM4."* (line 211)

*"Histograms of the mean and variance of the initial concentration over the 50 realizations are shown in Figure SM5 if a visualization of the range of values and of the variability is needed."* (line 238)

[Figure]

Figure SM6: Convergence of the mean and variance of the time required for 50% of the total recorded mass to reach a well.

[Figure]

Figure SM7: Convergence of the mean and variance of the concentration exceeded by 50% of the wells.

- Line 151-195: I see the detailed description of the estimation of the recharge rate to better fit in an Appendix.

  We would be open to moving this section to an appendix and replacing it with a short paragraph describing the core concept, if the other reviewers and associate editor agree.

- Line 230-233: *'The detailed discretization of the velocity field described above is capturing the most relevant characteristics affecting the macro-dispersive transport behavior (LaBolle, 1999; LaBolle and Fogg, 2001; Weissmann et al., 2002; Henri and Harter,2019). Therefore,*

*effects of grid-scale dispersion are assumed to be negligible.'*
So, does the Author set the tensor D in (10) equal to zero? Please clarify.

Yes, the dispersion tensor is equal to zero. We propose to clarify with the following text:

*"Therefore, effects of grid-scale dispersion are assumed to be negligible, i.e., **D=0** and Eq. 10 is simplified to $x_p(t + \Delta t) = x_p(t) + \Delta t. v(x_p(t))$. This assumption, which potentially impacts NPS management metrics, is further evaluated in Appendix B." (line 233)*

- Line 244-246: *'Three relevant nonpoint source (NPS) pollution management metrics are considered to measure the stochastic simulation outcomes: the probability distribution of pollutant travel times to wells, the probability distribution of pollutant con- centration in wells, and the probability distribution of source locations.'*
These are the 3 quantities of interest, why do you introduce them in the 2.3.1 Pollutant travel times section? Better just before.

Excellent observation. This sentence is now moved where you advised it should be.

- Line 250-251: *'Following a stochastic approach, probability density functions (pdfs) of travel times ti are obtained by determining the histogram of ti in 150 simulated wells'.*
Please note that the histogram is not a pdf, whereas the latter is associated with a continuous variable and the former to a discrete variable. I would limit to say to that the pdf are estimated on top of the 150 simulated wells. Furthermore, at lines 244-246 the probability distribution is mentioned, this is not the pdf. Also, at line 214 the authors say that they analyzed the probability distribution, but then in Figure 3 they depict the pdf. Please check for the consistency of the terminology/results through all the work.

Thank you for point this out. The terminology has been made consistent. For clarification, we do not analyze the *pdf* of concentrations but of travel times only. Also, a histogram can be obtained for both, discrete and continuous variables (e.g., velocity, concentration). It is a binned representation of an (unknown) pdf. Among others, the aforementioned sentence has been changed to:

*"Following a stochastic approach, probability density functions (pdfs) of travel times $t_i$ are obtained using time series from 150 simulated wells (50 realizations, each with 3 wells)." (line 250)*

- Lines 262-263: *'NPS pollution management may also require the assessment of the effective source area, i.e., the capture zone or contributing area of the pollution observed in a production well.'*
Please avoid to use source zone, this is typically used to indicate the area covered by the contaminant at the initial time (regardless if it reach the well or not), the capture zone of the well is way more clear as the wording in my opinion.

This terminology is, indeed, often confusing. Barlow et al. (2018) provide some good thoughts and clarification about the different terms used in the literature under different contexts (water budget, transport, …). The authors define "capture zone" as "the three-dimensional, volumetric portion of a groundwater flow field that discharges water to a well". This is not what we show in our work. We analyze the 2D projection of this so-defined capture zone, which correspond more to a source or contributing area.

*Reference*: Barlow, P. M., Leake, S. A. and Fienen, M. N. (2018), Capture Versus Capture Zones: Clarifying Terminology Related to Sources of Water to Wells. Groundwater, 56: 694-704. doi:10.1111/gwat.12661.

- Line 275-277: *'The NPS metrics from fully heterogeneous simulations are compared to the NPS metrics obtained from a range of upscaled, homogenized simulations that employ effective homogeneous properties rather than the original heterogeneous distribution of the K, r, and c0 terms.'*
Please consider avoiding the word upscale/upscaling (here and in the whole text), since this inherently implies a change of scale (e.g., from pore to continuum or from continuum to continuum) and it is typically associated with a change of the governing equations used to describe the process (e.g., effective model for the solute transport involving non-local terms) whereas here the Author conducted a simple homogenization (with the arithmetic average as a rule) of the diverse terms.

  We, in fact, employ a change of scale here: Instead of characterizing recharge, loading, or K at the local scale, we utilize a sub-regional scale "effective" value. Previous work refers to upscaling of the hydraulic conductivity as the estimation of effective parameters aimed to be used in regular flow equations (e.g., Fleckenstein and Fogg, 2008). In our work, we also estimated effective velocities and dispersivities in our homogenization. The terms upscaling and homogenization are both used regularly along the manuscript, which, we believe, makes our methodology clear.

  Reference: Fleckenstein, J.H., and Fogg, G.E. (2008). Efficient upscaling of hydraulic conductivity in heterogeneous alluvial aquifers. Hydrogeology Journal, 16:1239. https://doi.org/10.1007/s10040-008-0312-3.

- Line 286-292: *'To simulate flow and transport in an equivalent homogeneous, upscaled K conditions, we estimate the effective longitudinal and transverse vertical hydraulic conductivity, K\*x and K\*z , and dispersion, alpha\*L and alpha\*T. Effective parameters in the longitudinal direction (K\*x and alpha\*L) are determined from the first and second spatial moments of a plume resulting from an injection of mass in a vertical plane of width 3000.0 m and depth 50.0 m. The same approach is adopted to estimate the effective parameters in the transverse vertical direction (K\*z and alpha\*T) by injecting particles in a horizontal plane covering the entire top of the domain. No extraction is considered in both cases in order to capture the natural behavior of the plume.'*
What about the K\*y and dispersivity in the y direction? Furthermore, 'No extraction is considered in both cases in order to capture the natural behavior of the plume', but the ensuing macro-dispersion of a plume is influenced by the presence of pumping well (e.g., radial versus uniform flow conditions), please clarify/justify this aspect.

  Transverse longitudinal (y-direction) transport is negligible in our simulation setting (nonpoint source and no gradient in y).

  We estimate apparent parameters (velocity and dispersion) under "natural" conditions in order to only capture the impact of K heterogeneity on transport. Scenarios accounting for homogenized K simulate explicitly extractions.

- Lines 317-316: *'For all simulations, early mass travel times are within a range of 10 to 100 years with an expected value (highest probability) of 50 years (Figure 3a).'*
For a continuous variable is not possible to define the value with the highest probability (it is possible for a discrete variable). Furthermore, the expected value of a pdf does not always coincide with the value where the value of the pdf is the highest. Clarify.

  Thank you for the concern and clarification. The sentence has been changed to:

  *"For all simulations, early mass travel times are within a range of 10 to 100 years with a peak of probability of 50 years (Figure 3a)." (line 317)*

- Lines 346-347: *'Spatial variability in the recharge is responsible for somewhat more uncertainty in the exact delineation of the capture zone along its margins than what is captured by the homogenization of r.'*
  Is not r the symbol to indicate the recharge? I think r should be replaced with C0 or vice-versa. Furthermore, focusing on the impact of the homogenization of the recharge (compare Fig. 4a with Fig. 4c) I would say that the homogenization of the latter is associated with a less spatially extended capture zone w.r.t. to the case in which the recharge is treated as heterogeneous, this does not mean that the there is more uncertainty in the delineation of the capture zone. Furthermore, looking to the Fig. 4 I would say that capture zone is not well-delineated along its border, due to the low number of Monte Carlo realizations. Please check it and revise the sentence.

  > Yes, it should have been c_0, sorry for the typo. The dark blue area in Figure 4 is associated with very low probability to reach a well (Pw). In our setting, it may seem complex to reach a "smoother" delineation of the area defined by Pw values tending to 0.

  > Furthermore, despite that Pw>0 in these locations, we do not believe that it should be interpreted as an indicator of the extend of the capture zone in a management effort. On the other hand, we interpret a decrease of probability in the "hot spot", i.e., the source area characterized by the highest Pw values as an increase of uncertainty. It is now made clearer:

  > *"Spatial variability in the recharge is responsible for somewhat more uncertainty (i.e., a decrease of the highest $P_w$ values) in the exact delineation of the capture zone along its margins than what is captured by the homogenization of $c_0$." (line 346)*

- Lines 379-380: *'For an individual well, the results indicate that there is a 10% chance for nitrate concentrations to start to rise before 30 years, a 50% chance to rise no later than 50 years, and a 90% chance to rise before 70 years.'*
  Should not be after X years?

  > We do believe that before is right. In Figure 6, $P_{90}$ represents the concentrations exceeded with a 90% probability. These concentrations rise at t=70 years, while the concentrations exceeded with a 10% probability ($P_{10}$) rise earlier, at t=30 years. Since we analyze probabilities of exceedance, there is a 90% chance for well concentrations to rise before 70 years, and a 10% chance to rise earlier, before 30 years.

- Lines 402-404: *'Homogenizing only concentration also leads to an underprediction, by about 20%, of concentrations exceeded by either 90%, 50%, or 10% of well, relative to the fully stochastic land use treatment (green lines in Figure 6).'*
  Is green correct?

  > Thank for pointing out this typo. It is not green but blue.

  > *"Homogenizing only concentration also leads to an underprediction, by about 20%, of concentrations exceeded by either 90%, 50%, or 10% of well, relative to the fully stochastic land use treatment (compare blue and red lines in Figure 6)." (line 402)*

- Lines 471-472: *'... only adds a moderate degree of uncertainty to the capture zone delineation...'*
  see Comment 15.

  > Here again, we associate uncertainty to a decrease of the highest probabilities to reach a well ($P_w$), and not with an increase of the surface covered by very low $P_w$ values. It is now made clearer in the paper.

- Line 480: *'The observed gradient of travel times'*
  I don't see any gradient (i.e.,∂ˊLGˇ) of the travel times evaluated by the Authors. There is a spatial variation of the mean travel time in Fig. 9, but this is different from a proper evaluation of the gradient. Please revise the sentence

    The sentence has been modified to:

    *"The observed spatial variation of the mean travel times, increasing with the distance from a well, is overestimated when $K$-fields are homogenized." (line 480)*

- Lines 487-489: *'Thus, contaminant mass reaching the top of the well has little variability – here only to the degree that the homogenization is done individually for each realization, leading to some minor variability in the homogenized K between realizations.'*
  Why does the fact that the mass reaching the top of the well exhibits a low variability (note that the Authors do not provide a quantification of it) lead to have minor variability in the homogenized K? Please clarify.

    This sentence has been reworked in light of your following remark. Please see below.

- Lines 489-491: *'More uncertainty is observed on the upstream side of the capture zone since it represents mass reaching the bottom of the screen, the vertical position of which is realization dependent'*
  I suppose this comment is related to Fig. 9, which depicts the expected value of the travel time to the well. The latter gradually increases as we move upstream w.r.t. to the well location. I don't see this as a measure of an increasing level of uncertainty! It could be that the expected value of the travel time increases, but it can also be that spread (e.g., measured through the variance) of the pdf decreases. Please consider evaluating at least the variance of the pdf of travel times as a quantification of the degree of uncertainty.

    Thank you for the comment. Indeed, Figure 9 does not display an estimation of travel time uncertainty. This paragraph discussing variability focused more on Figure 8 displaying the probabilities to reach a well. This section has been moved earlier in the section and the appropriate Figure has been referred to in order to avoid confusion.

    Given the already extensive length of the manuscript, we believe that it is not necessary to add a more detailed discussion about the uncertainty of travel times associated to any location of the capture zone. Here, we focus more a more practical aspect of the problem, i.e., the assessment of the performance of a homogenization of K in reproducing expected travel times.

    For your information, our previous paper (Henri and Harter, 2019) discusses the uncertainty of travel times associated to any location of the capture zone in similar conditions.

    Reference: Henri, C., & Harter, T. (2019). Stochastic assessment of nonpoint source contamination: Joint impact of aquifer heterogeneity and well characteristics on management metrics. Water Resources Research, 55. https://doi.org/10.1029/ 2018WR024230.

- Lines 520-521: *'Results show that homogenized K-fields perform more poorly to predict the lowest concentrations (P90) than the highest ones (P50 and P10).'*
  Why P90 is associated with low concentrations? P90 is a probability, looking at Fig. 10a I see that for a given concentration (either high or low) there is a time necessary to exceed this level of concentration in the well with a probability of 0.9. The way I interpret Fig. 10 is that homogenized solutions are in good agreement with the heterogeneous case, when it is a matter of evaluating low (P10) and middle (P50) probability of exceedance of a given

concentration, but the homogenized solutions do not work well for high (P90) probability of exceedance of whatever concentration.

Thank you for raising out this point. This paragraph was more intended to be an interpretation of what we call a "regional" analysis.

$P_{90}$ represents, indeed, a high (90%) probability of exceedance. In other words, only there is only 10% of chance that a well will present a lower concentration. In what we describe as a regional analysis, this means that 10% of wells present concentration lower than the one shown in $P_{90}$. We believe that it a fair assessment to qualify these concentrations as the lowest concentrations.

The paragraph has been moved into the appropriate section.

- Conclusion: I note a change of style in the conclusion, whereas there is a more consistent and proper use of the terminology with respect to the rest of the work.

Thank you.

---

## Author Response (AR1)

**Editor Decision: Publish subject to minor revisions**

**Comments to the Author:**

The reviews offer detailed points that require consideration, with appropriate clarifications in a revised manuscript.

The authors' responses appear well thought out and convincing, and a revised manuscript reflecting the additional information and clarifications can fully address the reviews. Please proceed with a thorough revision.

I note that some of the author responses (particularly to referee #2) consisted of a detailed commentary, but actual revision to the manuscript was not included. Please reconsider - in most cases, where a detailed response is needed, even to justify/clarify why something was or was not included in the analysis, at least a short additional explanation is helpful in the manuscript itself (if a referee had a question, it's reasonable to assume that other readers may have the same one).

Thank you for your comments and for inviting us to submit a revised manuscript.

Indeed, a few points discussed while answering the referee #2 deserved an additional clarification in the manuscript. We took in consideration your point, and all concerns of both reviewers are now addressed in the new version of the paper. These added sentences are listed in our point-by-point response to the reviews that you can find below. Changes made in the manuscript are listed in green. A marked-up manuscript version is also provided.

**Referee #1 Evaluations**

**Overview**

This is an interesting work that brings together many contributions in the field of probabilistic risk assessment (PRA) in aquifers to investigate transport of non-point sources (NPS). The authors explore parameters such as recharge rates and contaminant loadings in the final model output. Furthermore, the authors attempt to reduce the complexity of the model by upscaling a set of spatially/temporally variable quantities, such as the hydraulic conductivity, on the management of NPS. Through the use of numerical simulations, the authors provide an analysis that couples vadose zone, aquifers and land use in a single framework.

I enjoyed reading this paper given that it aims in bringing in elements of stochastic hydrogeology into decision making. The material is well written and organized. The illustrations are clear and well depicted. The referencing is also appropriate although some contributions in the PRA of contaminated aquifers are missing. This is not a big issue. Through the use of scaling arguments (i.e. compliance planes, source sizes etc) the authors make a compelling argument to evoke upscaling for the problem at hand. They claim that due to significant mixing in the compliance plane and the lack of significant variability in NPS solutes, the uncertainty in predictions are reduced thus leading more simplified approaches for modeling such complex systems. Results indicate that the mass arrival time distributions are not that sensitive to the spatial variability of recharge and solute loading whereas some sensitivity is observed for the concentration signal and capture zone estimation. The authors also show that homogenization of the conductivity affects the uncertainty of arrival times.

Thank you for your overall positive evaluation of the manuscript and for your interest. We hope that our answers to your specific comments will clarify your concerns.

**Specific comments**

• The authors refer to the word ergodicity multiple times. Ergodicity in what? I think they are referring to ergodicity in the transport behavior. If so, provide a quantitative measure of what ergodicity is. For example, the ratio between the source zone dimension and correlation scale needs to be large. If this is the case, then why one would need to quantify uncertainty due to the conductivity field? The spatial statistics is representative of the ensemble statistics. This needs to be better discussed.

See response after the next bullet.

 I am not sure if I missed this in the text but it would be interesting to see if the upscaled dispersion reaches its Fickian limit. Looking at figure SM6, it seems that this is not the case and therefore, transport is still subject to uncertainty. To my understanding, based on the histograms, these upscaled dispersion coefficients reported in figure SM6 are not the ones in the Fickian limit and therefore ergodicity is not attained. So how is it that the authors claim "ergodicity" in this paper?

Thank you for raising your concern about ergodicity in these two comments. The context in which the term was used in the manuscript was, indeed, unclear.

The assumption of ergodicity is here employed NOT in the sense that we assume the NPS plume to be so large that there is no variability in plume moments between realizations. Instead, we here employ the ergodicity hypothesis in the same way as described, e.g., by Rajaram (2002) or Gelhar (1993). There, a single large realization (spatially extending over many correlation scales) of the K field is used to compute sample moments (e.g., of head,

concentration). The ergodic hypothesis is employed to justify comparing these sample moments to the theoretical ensemble moments obtained analytically. But the ensemble moments also characterize the uncertainty about, e.g., head or concentration at a single unmeasured location within the aquifer represented by the simulation.

Analogous, we here use the ergodic hypothesis to postulate that the statistical sampling across the 150 well and 150 source areas simulated is representative of the ensemble moments (the stochastic management metrics) at an individual unmeasured well and its source area in the real-world aquifer represented. The 150 simulated wells and source areas are analogous to the single large realization in Rajaram (2002), since the K field is stationary.

In other words, we must assume ergodicity and stationarity to be able to equate the stochastic NPS management metrics obtained from across 150 samples (wells, source areas) to their ensemble properties. The ensemble properties in turn apply to any (unmeasured) well and its source area in the aquifer system. They characterize the uncertainty about these metrics at each (unmeasured) location (well, source area).

We propose to clarify this point in the manuscript (see below) and also refer to the histogram of the mean and variance of the K fields in supplementary material (Figure SM3, to be updated in the new version of the manuscript and shown below) showing stationarity in the moments of the K field, i.e., a very narrow range of values for the mean and variance, and therefore displaying that structural ergodicity can be assumed.

"Then, stochastic management metrics quantify both, the mean and variability of pollution levels across a large sample of production wells encountered over a basin as well as the expected value and uncertainty about pollution levels at an individual well. This is done by simulating stationary random fields (Figure SM3) and assuming ergodic conditions [e.g., Gelhar, 1993; Rajaram, 2002]." (line 122)

Reference: Rajaram (2002), Perturbation theories for the estimation of macrodispersivities in heterogeneous aquifers, in Rao S. Govindaraju, Stochastic methods in subsurface contamination hydrology, p13-62.

Gelhar, L. W. (1993), Stochastic subsurface hydrology: Englewood Cliffs, New Jersey, Prentice-Hall, 390 p.

Figure SM3: Histogram of the mean (top) and variance (bottom) of the lognormal hydraulic conductivity.

• It would be interesting to see how the conclusions regarding recharge reported in this paper compare with the ones reported in the works of Rubin and Bellin (1994) WRR and Li and Graham (1999) WRR. These authors investigate the impact of recharge and its randomness on travel time pdfs.

Thank you for drawing our attention to these relevant papers. First, it might be interesting to note the differences between our simulation setup and the one used in Rubin and Bellin (1994) and Li and Graham (1999): The two aforementioned papers are (semi-) analytically analyzing 2D transport, with a point source and no pumping. Rubin and Bellin (1994) assumes uniform recharge and found that recharge increases longitudinal plume spreading. Li and Graham (1999) do account for spatial variability in recharge further increases longitudinal spreading, and that uncertainty in the recharge spatial variability increases the uncertainty in solute concentrations.

On the other hand, we here simulate 3D transport, with non-point source transport, and significant pumping. Our results show little effect of recharge spatial variability (correlated to the top of the K-field) on travel times, while homogenization of recharge leads to increased uncertainty in well concentrations.

Results highlight that a potential superficial increase in lateral spreading is less (or non-) relevant in a 3D nonpoint source setting as indicated by our analysis of travel time statistics.

Concerning concentration statistics, the difference in outputs between Li and Graham (1999) and our work could indicate that decorrelated spatial variability of the recharge rate adds uncertainty, while a r-K correlation increases the conditioning of the flow field, which leads to decreased uncertainty. We recall that in our setting, the existence of a r-K correlation is a result of the explicit simulation of water flow in the unsaturated area.

In addition, considering three dimensions, as well as the presence of extraction wells adds some degree of complexity in transport and uncertainty propagation. Fully understanding these processes would require a significant effort of its own.

We propose to discuss these two points in a new version of the manuscript:

"Previously, Li and Graham (1998) investigated the impact of recharge spatial variability in a more theoretical and simplified 2D heterogeneous aquifer contaminated by a point source under non-pumping conditions. The work highlights that spatial variability in recharge increases spreading, especially in the transverse direction. In our 3D NPS setting, transverse spreading is less relevant (Figure 3) and we do not observe the increase in variability." (line 327)

"Li and Graham (1998) stochastically analyze the impact of spatially random recharge rate on transport in a 2D point source setting. Their work concluded that, for those conditions, large variability in – and therefore uncertainty about - recharge increases uncertainty in solute concentration. In our work, we observe the opposite. The difference may be partly due to the 3D non-point source transport, and partly caused by the implicit correlation between the hydraulic conductivity and the recharge rate in our scenarios, which may increase the conditioning of the flow field that leads to the observed decrease of uncertainty relative to the homogenized scenario." (line 401)

• Line 460: "The results here confirm that..., but also put the macro-dispersive process. . .". I could not understand the meaning of this sentence. Please revise its structure. Thanks.

Thank you for pointing it out. This sentence has been changed to:

"The results presented here confirm this observation for the case of non-point source contaminations, but also highlight the generation of a quasi-macro-dispersive process through the (vertical) well mixing process." (line 460)

**Referee #2 Evaluations**

The present work deals with the feasibility of simplify the conceptual modelling of non- point source (NPS) contamination of an aquifer, in the presence of an active pumping well. The simplification consists of spatial homogenization of otherwise heterogeneous terms (recharge rate and contaminant concentration of the recharging water) and of the aquifer hydraulic properties (conductivity). I liked the general goal/purpose of the study (even though there is no theoretical/technical novelty in the employed method- ology) and its practically oriented nature. Nevertheless, it is my opinion that there are several unclear points and unstained observations that should be addressed before acceptance for HESS. I list my concerns below. I do see a very good paper after addressing these issues/confusions. I also hope that my comments help in making the paper shorter and clearer, it becomes quite hard to follow it from the beginning to the end.

Thank you for your constructive and well-thought-out comments. We appreciate the attention to the details of our work. Please see below our answers (in blue) to your concerns. Proposed additions to, deletions from, or modification of the manuscript are written in green.

Line 5-9: 'On the other hand, concentration levels of some key NPS contaminants (salinity, nitrate) vary within a limited range (

Figure SM3: Histogram of the mean (top) and variance (bottom) of the lognormal hydraulic conductivity.

Reference: Rajaram (2002), Perturbation theories for the estimation of macrodispersivities in heterogeneous aquifers, in Rao S. Govindaraju, Stochastic methods in subsurface contamination hydrology, p13-62.

Gelhar, L. W. (1993), Stochastic subsurface hydrology: Englewood Cliffs, New Jersey, Prentice-Hall, 390 p.

The Authors named the regional analysis ('pollution variability across an ensemble of production wells encountered over a basin') (see also Section 6) the scenario in which both the conductivity and the land crop usage varies between the Monte Carlo simulations. They referred to the single well analysis ('uncertainty about pollution levels at an individual well') when only the conductivity varies among the MC simulations (with the crop arrangement fixed). To me this distinction is not meaningful, since it implies that in the groundwater basin there are sub-portion (the simulated domain) subjected to the very same boundary conditions (aside from the infiltration rate) and that these sub-portions do not influence each other (they are far away from one another). In my vision the Authors have, given a domain of interest and deterministic initial and boundary conditions (aside from the infiltration rate), conducted the uncertainty analysis (i) for a given (i.e., conditional) to a crop arrangement and (ii) considering the uncertainty in the crop arrangement, proven then that the uncertainty in the latter is not an influential factor (i.e., there is ergodicity w.r.t. to the crop usage). Please consider this aspect, the regional and one-well distinction seems to me confusing and not well supported by the investigated set-up.

Indeed, we assume that there are sub-portions of the aquifer that do not influence each other. The size of the simulation domain was chosen to fully accommodate the source area of three production wells. Since we show that their source areas do not overlap, each well's area of capture can indeed be considered independent. This point is now explicitly mentioned in the manuscript:

"To assess the spatial variability of NPS management metrics across an ensemble of well locations in a groundwater basin, the equiprobable realizations of the aquifer system represent the variety of locations across a basin with geostatistically similar geological features. This is true since the domain is designed to ensure that source areas of the three production wells are fully accommodated and that each well's area of capture can be considered independent. In the case of a regional analysis, land-use is simulated as a random process." (line 120)

The reviewer is also correct that we make a conceptual distinction between the case of assessing the uncertainty about an individual well's source area and contaminant travel time, and the case of assessing the variability of source area and contaminant travel time of many wells in a region with a stationary hydraulic conductivity, recharge, NPS mass loading, and landuse pattern. We argue that this is an important theoretical distinction, for the same reason that the concept of ergodicity is relevant.

From a theoretical standpoint, therefore, generating 50 realizations (each with 3 wells), without changing actual landuse is, in principal, only assessing the uncertainty about recharge, loading, and K under a given spatial configuration of landuse (which would be specific to each of the three wells).

Then also making landuse spatial pattern a stochastic variable is conceptually consistent with the idea of simulating 150 different wells, each with its own spatial landuse arrangement.

But the reviewer is correct in pointing out that there is negligible difference between these two, effectively proving ergodicity w.r.t. landuse. But we do not believe it is useful to say this explicitly (e.g., in a sentence at the end of section 6), because we don't want to confuse the reader with the "other" (valid) interpretation of ergodicity that the reviewer is in fact employing here.

We therefore propose to add the word "pattern" to the line 533, make a small change in line 536

"To adapt the simulation setup to the regional stochastic analysis, the spatial distribution pattern of crops...." (line 533)

"..... interpretation of the stochastic results, the range OF individual travel times, capture zones, ...." (line 536, capitalized word to be inserted)

Line 115-117: 'Assuming ergodicity (Dagan, 1990), stochastic analysis is applied to first quantify uncertainty about pollution outcomes at individual wells and to secondly quantify regional spatial variability in pollution outcomes across an ensemble of wells.' Please revise this sentence according with Comment 2-3. Furthermore, in the way it is written it means that ergodicity is needed in order to quantify uncertainty (this is the case for example in geostatistical approach where spatially distributed measure of conductivity in a field, i.e., in a single realization, are employed to describe the ensemble statistic of conductivity) in this study, i.e., that automatically a single realization is sufficient to describe the behavior of the ensemble, whereas, it is my understanding that the Author did the other way around: prove the validity of ergodicity (for the investigated quantity) w.r.t. to the crop usage. If ergodicity was originally assumed, no need to do many simulations.

See our answer to the previous two comments.

Line 129-130: 'The histograms of the mean and the variance of the logarithm of K are shown in Figure SM3. Fifty realizations were sufficient to converge the lower statistical moments of K and of the resulting mean velocities (Figure SM7)'.

It is not clear at all that the Authors are referring to the spatial mean and spatial variance (this is my impression) evaluated for each field of K, and then doing the histogram of these quantity. Is it so? Why do we care about it? How do the Author prove the convergence of these spatial mean and spatial variance employing 50 realizations? To which values should these quantities converge? I am way more concerned about: are the 50 realizations enough to ensure the convergence of the statistics (e.g., pdf, CDF) of the output quantities of interest (i.e., travel time, breakthrough curves at the well and capture zone)? This is aspect is not investigated at all by the Authors and looking to the high spatial variance of K (from 10 to 18 approximately) I am afraid that 50 realizations are not enough, even if the source is spatially distributed. Please analyze the convergence of the investigated results w.r.t. to the Monte Carlo simulations. Furthermore, regarding the histogram in SM3 and SM4 for the recharge rate and the concentration of pollutant in the recharge, why do we care about them? Please clarify.

As mentioned before, the histogram of the mean and variance of the InK fields were erroneous (they were considering the first layer of the domain only). Please apologize the mistake. Below are the corrected histograms, which provide a better measure of the high degree to which the entire range of K variability is captured in each of the 50 realizations.

The convergence of the apparent velocities and of apparent dispersivities, which are derived from particle plumes, give an indication about convergence of transport metrics. The dataset used to compute statistics is composed of 150 values of well concentrations and of travel times (3 wells per realization). For the sake of clarity, we now also show in Figure SM6 and SM7 (see below) the convergence of the mean and variance of the travel times and of the 50% highest concentrations, respectively. We propose to add the following text to the manuscript:

"Figure SM6 shows satisfactorily the convergence of the mean and variance of  $t_{50}$ ." (line 252)

*"Figure SM7 shows satisfactorily the convergence of the mean and variance of the concentration exceeded by 50% of wells." (line 259)*

The histograms for the recharge rate (SM4) and the input concentrations (SM5) are given in supplementary material in case a reader may have a concern about the range of values used. The following text would be added to the manuscript:

"For indications about the range of values and degree of variability, histograms of the mean and variance of the recharge rates applied in the 50 realizations of heterogeneous cases are shown in Figure SM4." (line 211)

"Histograms of the mean and variance of the initial concentration over the 50 realizations are shown in Figure SM5 if a visualization of the range of values and of the variability is needed." (line 238)

Figure SM6: Convergence of the mean and variance of the time required for 50% of the total recorded mass to reach a well.

Figure SM7: Convergence of the mean and variance of the concentration exceeded by 50% of the wells.

• Line 151-195: I see the detailed description of the estimation of the recharge rate to better fit in an Appendix.

We would be open to moving this section to an appendix and replacing it with a short paragraph describing the core concept, if the other reviewers and associate editor agree.

Line 230-233: 'The detailed discretization of the velocity field described above is capturing the most relevant characteristics affecting the macro-dispersive transport behavior (LaBolle, 1999; LaBolle and Fogg, 2001; Weissmann et al., 2002; Henri and Harter, 2019). Therefore, effects of grid-scale dispersion are assumed to be negligible.' So, does the Author set the tensor D in (10) equal to zero? Please clarify.

Yes, the dispersion tensor is equal to zero. We propose to clarify with the following text:

"Therefore, effects of grid-scale dispersion are assumed to be negligible, i.e., **D**=0 and Eq. 10 is simplified to  $x_p(t + \Delta t) = x_p(t) + \Delta t. v(x_p(t))$ . This assumption, which potentially impacts NPS management metrics, is further evaluated in Appendix B." (line 233)

Line 244-246: 'Three relevant nonpoint source (NPS) pollution management metrics are considered to measure the stochastic simulation outcomes: the probability distribution of pollutant travel times to wells, the probability distribution of pollutant con- centration in wells, and the probability distribution of source locations.'

These are the 3 quantities of interest, why do you introduce them in the 2.3.1 Pollutant travel times section? Better just before.

Excellent observation. This sentence is now moved where you advised it should be.

Line 250-251: 'Following a stochastic approach, probability density functions (pdfs) of travel times ti are obtained by determining the histogram of ti in 150 simulated wells'. Please note that the histogram is not a pdf, whereas the latter is associated with a continuous variable and the former to a discrete variable. I would limit to say to that the pdf are estimated on top of the 150 simulated wells. Furthermore, at lines 244-246 the probability distribution is mentioned, this is not the pdf. Also, at line 214 the authors say that they analyzed the probability distribution, but then in Figure 3 they depict the pdf. Please check for the consistency of the terminology/results through all the work.

Thank you for point this out. The terminology has been made consistent. For clarification, we do not analyze the *pdf* of concentrations but of travel times only. Also, a histogram can be obtained for both, discrete and continuous variables (e.g., velocity, concentration). It is a binned representation of an (unknown) pdf. Among others, the aforementioned sentence has been changed to:

"Following a stochastic approach, probability density functions (pdfs) of travel times  $t_i$  are obtained using time series from 150 simulated wells (50 realizations, each with 3 wells)." (line 250)

• Lines 262-263: 'NPS pollution management may also require the assessment of the effective source area, i.e., the capture zone or contributing area of the pollution observed in a production well.'

Please avoid to use source zone, this is typically used to indicate the area covered by the contaminant at the initial time (regardless if it reach the well or not), the capture zone of the well is way more clear as the wording in my opinion.

This terminology is, indeed, often confusing. Barlow et al. (2018) provide some good thoughts and clarification about the different terms used in the literature under different contexts (water budget, transport, ...). The authors define "capture zone" as "the threedimensional, volumetric portion of a groundwater flow field that discharges water to a well". This is not what we show in our work. We analyze the 2D projection of this so-defined capture zone, which would correspond more to a source or contributing area. However, we do understand that both terms (source area and capture zone) has been used in the literature to designate this 2D projection. In order to avoid an eventual confusion, the new manuscript acknowledges this fact by providing the abovementioned reference and by explicitly mentioning that these terms have equivalent meaning throughout the paper:

"Important aspects of NPS pollution are pollutant travel times, the location of well source areas (also known as capture zones; Barlow et al. 2018) to identify specific pollution sources, and the long-term evolution of contaminant levels in and across affected wells and streams." (line 23)

"The stochastic capture zone (or source area) is the area characterized by Pw(x,y)>0." (line 336)

*Reference*: Barlow, P. M., Leake, S. A. and Fienen, M. N. (2018), Capture Versus Capture Zones: Clarifying Terminology Related to Sources of Water to Wells. Groundwater, 56: 694-704. doi:10.1111/gwat.12661.

Line 275-277: 'The NPS metrics from fully heterogeneous simulations are compared to the NPS metrics obtained from a range of upscaled, homogenized simulations that employ effective homogeneous properties rather than the original heterogeneous distribution of the *K*, *r*, and c0 terms.'

Please consider avoiding the word upscale/upscaling (here and in the whole text), since this inherently implies a change of scale (e.g., from pore to continuum or from continuum to continuum) and it is typically associated with a change of the governing equations used to describe the process (e.g., effective model for the solute transport involving non-local terms) whereas here the Author conducted a simple homogenization (with the arithmetic average as a rule) of the diverse terms.

We, in fact, employ a change of scale here: Instead of characterizing recharge, loading, or K at the local scale, we utilize a sub-regional scale "effective" value. Previous work refers to upscaling of the hydraulic conductivity as the estimation of effective parameters aimed to be used in regular flow equations (e.g., Fleckenstein and Fogg, 2008). In our work, we also estimated effective velocities and dispersivities in our homogenization. The terms upscaling and homogenization are both used regularly along the manuscript, which, we believe, makes our methodology clear.

In order to clarify the meaning of this term, the new manuscript refers to the abovementioned paper:

"The NPS metrics from fully heterogeneous simulations are compared to the NPS metrics obtained from a range of upscaled (e.g., Fleckenstein and Fogg, 2008), homogenized simulations that employ effective homogeneous properties rather than the original heterogenous distribution of the K..." (line 275)

Reference: Fleckenstein, J.H., and Fogg, G.E. (2008). Efficient upscaling of hydraulic conductivity in heterogeneous alluvial aquifers. Hydrogeology Journal, 16:1239. https://doi.org/10.1007/s10040-008-0312-3.

Line 286-292: 'To simulate flow and transport in an equivalent homogeneous, upscaled K conditions, we estimate the effective longitudinal and transverse vertical hydraulic conductivity, K\*x and K\*z , and dispersion, alpha\*L and alpha\*T. Effective parameters in the

longitudinal direction (K\*x and alpha\*L) are determined from the first and second spatial moments of a plume resulting from an injection of mass in a vertical plane of width 3000.0 m and depth 50.0 m. The same approach is adopted to estimate the effective parameters in the transverse vertical direction (K\*z and alpha\*T) by injecting particles in a horizontal plane covering the entire top of the domain. No extraction is considered in both cases in order to capture the natural behavior of the plume.'

What about the K\*y and dispersivity in the y direction? Furthermore, 'No extraction is considered in both cases in order to capture the natural behavior of the plume', but the ensuing macro-dispersion of a plume is influenced by the presence of pumping well (e.g., radial versus uniform flow conditions), please clarify/justify this aspect.

Transverse longitudinal (y-direction) transport is negligible in our simulation setting (nonpoint source and no gradient in y).

We estimate apparent parameters (velocity and dispersion) under "natural" conditions in order to only capture the impact of K heterogeneity on transport. Scenarios accounting for homogenized K simulate explicitly extractions.

*"The transverse horizontal (y-direction) component of transport is considered negligible given the size of the NPS plume and given that no gradient in y was applied." (line 288)*

Lines 317-316: 'For all simulations, early mass travel times are within a range of 10 to 100 years with an expected value (highest probability) of 50 years (Figure 3a).'
 For a continuous variable is not possible to define the value with the highest probability (it is possible for a discrete variable). Furthermore, the expected value of a pdf does not always coincide with the value where the value of the pdf is the highest. Clarify.

Thank you for the concern and clarification. The sentence has been changed to:

"For all simulations, early mass travel times are within a range of 10 to 100 years with a peak of probability of 50 years (Figure 3a)." (line 317)

Lines 346-347: 'Spatial variability in the recharge is responsible for somewhat more uncertainty in the exact delineation of the capture zone along its margins than what is captured by the homogenization of *r*.'

Is not r the symbol to indicate the recharge? I think r should be replaced with C0 or viceversa. Furthermore, focusing on the impact of the homogenization of the recharge (compare Fig. 4a with Fig. 4c) I would say that the homogenization of the latter is associated with a less spatially extended capture zone w.r.t. to the case in which the recharge is treated as heterogeneous, this does not mean that the there is more uncertainty in the delineation of the capture zone. Furthermore, looking to the Fig. 4 I would say that capture zone is not well-delineated along its border, due to the low number of Monte Carlo realizations. Please check it and revise the sentence.

Yes, it should have been c\_0, sorry for the typo. The dark blue area in Figure 4 is associated with very low probability to reach a well (Pw). In our setting, it may seem complex to reach a "smoother" delineation of the area defined by Pw values tending to 0.

Furthermore, despite that Pw>0 in these locations, we do not believe that it should be interpreted as an indicator of the extend of the capture zone in a management effort. On the other hand, we interpret a decrease of probability in the "hot spot", i.e., the source area characterized by the highest Pw values as an increase of uncertainty. It is now made clearer:

"Spatial variability in the recharge is responsible for somewhat more uncertainty (i.e., a decrease of the highest  $P_w$  values) in the exact delineation of the capture zone along its margins than what is captured by the homogenization of  $c_0$ ." (line 346)

Lines 379-380: 'For an individual well, the results indicate that there is a 10% chance for nitrate concentrations to start to rise before 30 years, a 50% chance to rise no later than 50 years, and a 90% chance to rise before 70 years.' Should not be after X years?

We do believe that before is right. In Figure 6,  $P_{90}$  represents the concentrations exceeded with a 90% probability. These concentrations rise at t=70 years, while the concentrations exceeded with a 10% probability ( $P_{10}$ ) rise earlier, at t=30 years. Since we analyze probabilities of exceedance, there is a 90% chance for well concentrations to rise before 70 years, and a 10% chance to rise earlier, before 30 years.

Lines 402-404: 'Homogenizing only concentration also leads to an underprediction, by about 20%, of concentrations exceeded by either 90%, 50%, or 10% of well, relative to the fully stochastic land use treatment (green lines in Figure 6).' Is green correct?

Thank for pointing out this typo. It is not green but blue.

"Homogenizing only concentration also leads to an underprediction, by about 20%, of concentrations exceeded by either 90%, 50%, or 10% of well, relative to the fully stochastic land use treatment (compare blue and red lines in Figure 6)." (line 402)

 Lines 471-472: '... only adds a moderate degree of uncertainty to the capture zone delineation...'
 Commont 15

see Comment 15.

Here again, we associate uncertainty to a decrease of the highest probabilities to reach a well ( $P_w$ ), and not with an increase of the surface covered by very low  $P_w$  values. It is now made clearer earlier in the paper (*line 346*).

• Line 480: 'The observed gradient of travel times'

I don't see any gradient (i.e.,â LG ) of the travel times evaluated by the Authors. There is a spatial variation of the mean travel time in Fig. 9, but this is different from a proper evaluation of the gradient. Please revise the sentence

The sentence has been modified to:

"The observed spatial variation of the mean travel times, increasing with the distance from a well, is overestimated when K-fields are homogenized." (line 480)

 Lines 487-489: 'Thus, contaminant mass reaching the top of the well has little variability – here only to the degree that the homogenization is done individually for each realization, leading to some minor variability in the homogenized K between realizations.' Why does the fact that the mass reaching the top of the well exhibits a low variability (note that the Authors do not provide a quantification of it) lead to have minor variability in the homogenized K? Please clarify.

This sentence has been reworked in light of your following remark. Please see below.

• Lines 489-491: 'More uncertainty is observed on the upstream side of the capture zone since it represents mass reaching the bottom of the screen, the vertical position of which is

**realization dependent'**

I suppose this comment is related to Fig. 9, which depicts the expected value of the travel time to the well. The latter gradually increases as we move upstream w.r.t. to the well location. I don't see this as a measure of an increasing level of uncertainty! It could be that the expected value of the travel time increases, but it can also be that spread (e.g., measured through the variance) of the pdf decreases. Please consider evaluating at least the variance of the pdf of travel times as a quantification of the degree of uncertainty.

Thank you for the comment. Indeed, Figure 9 does not display an estimation of travel time uncertainty. This paragraph discussing variability focused more on Figure 8 displaying the probabilities to reach a well. The phrase has been reworked and the appropriate Figure has been referred to in order to avoid confusion.

"Thus, contaminant mass reaching the top of the well has little variability - here only to the degree that the homogenization is done individually for each realization - leading to some minor realization-to-realization variability at the downstream side of the capture zone for the homogenized K (Figure 8). More uncertainty is observed on the upstream side of the capture zone since it represents mass reaching the bottom of the screen, the vertical position of which is realization dependent." (line 486)

Given the already extensive length of the manuscript, we believe that it is not necessary to add a more detailed discussion about the uncertainty of travel times associated to any location of the capture zone. Here, we focus more a more practical aspect of the problem, i.e., the assessment of the performance of a homogenization of K in reproducing expected travel times.

For your information, our previous paper (Henri and Harter, 2019) discusses the uncertainty of travel times associated to any location of the capture zone in similar conditions.

Reference: Henri, C., & Harter, T. (2019). Stochastic assessment of nonpoint source contamination: Joint impact of aquifer heterogeneity and well characteristics on management metrics. Water Resources Research, 55. https://doi.org/10.1029/ 2018WR024230.

Lines 520-521: 'Results show that homogenized K-fields perform more poorly to predict the lowest concentrations (P90) than the highest ones (P50 and P10).' Why P90 is associated with low concentrations? P90 is a probability, looking at Fig. 10a I

see that for a given concentration (either high or low) there is a time necessary to exceed this level of concentration in the well with a probability of 0.9. The way I interpret Fig. 10 is that homogenized solutions are in good agreement with the heterogeneous case, when it is a matter of evaluating low (P10) and middle (P50) probability of exceedance of a given concentration, but the homogenized solutions do not work well for high (P90) probability of exceedance of whatever concentration.

Thank you for raising out this point. This paragraph was more intended to be an interpretation of what we call a "regional" analysis.

 $P_{90}$  represents, indeed, a high (90%) probability of exceedance. In other words, only there is only 10% of chance that a well will present a lower concentration. In what we describe as a regional analysis, this means that 10% of wells present concentration lower than the one shown in  $P_{90}$ . We believe that it a fair assessment to qualify these concentrations as the lowest concentrations.

The paragraph has been moved into the appropriate section.

"For instance, results show that homogenized K-fields perform more poorly to predict the lowest concentrations (P90) than the highest ones (P50 and P10). From a NPS pollution management perspective, the accuracy of the higher concentrations exceeded by half of wells or even by just 10% of wells is most critical, since they are more likely to exceed the MCL. The homogenized predictions are least accurate during the transition (breakthrough) period when concentrations in the vertically mixed sample obtained from a well are strongly controlled by travel time pdfs, which in turn are affected by the heterogeneity in the land use and aquifer dynamics." (line 552)

Conclusion: I note a change of style in the conclusion, whereas there is a more consistent and proper use of the terminology with respect to the rest of the work.

Thank you.

•

**On the Conceptual Complexity of Non-Point Source Management: Impact of Spatial Variability**

Christopher Vincent Henri1, Thomas Harter1, and Efstathios Diamantopoulos2

1University of California, Davis, Center for Watershed Sciences, Veihmeyer Hall, Davis, CA 95616, USA. 2University of Copenhagen, Department of Plant and Environmental Sciences, Thorvaldsensvej 40, Copenhagen, DK-1871, Denmark.

Correspondence: Christopher Vincent Henri (chenri@ucdavis.edu)

**Abstract.** Non-point source (NPS) pollution has degraded groundwater quality of unconsolidated sedimentary basins over many decades. Properly conceptualizing NPS pollution from the well scale to the regional scale leads to complex and expensive numerical models: Key controlling factors of NPS pollution - recharge rate, leakage of pollutants, and soil and aquifer hydraulic properties - are spatially and, for recharge and pollutant leakage, temporally variable. This leads to high uncertainty in

- 5 predicting well pollution. On the other hand, concentration levels of some key NPS contaminants (salinity, nitrate) vary within a limited range (